# Sensor Monitoring of Conveyor Working Operation with Oscillating Trough Movement

**DOI:** 10.3390/s25082466

**Published:** 2025-04-14

**Authors:** Leopold Hrabovský, Štěpán Pravda, Martin Fries

**Affiliations:** Faculty of Mechanical Engineering, Department of Machine and Industrial Design, VSB—Technical University of Ostrava, 17. listopadu 2172/15, 708 00 Ostrava, Czech Republic

**Keywords:** vibrating conveyor, harmonic vibration, effective speed value, sensor vibration measurement, circular motion of trough

## Abstract

This paper presents measured vibration magnitudes on the trough surface and on the frame of a laboratory model of a vibrating conveyor, detected by acceleration sensors. The vibration source is a DC asynchronous vibration motor with two discs with unbalanced masses mechanically attached to the end parts of the rotor. The trough of the vibrating conveyor is supported by four rubber springs of two types, which are characterised by considerable spring stiffness. Digital signals were recorded using the DEWESoft SIRIUSi measuring apparatus, which carries information about the magnitude of acting vibrations, which can be remotely transmitted from their place of action via a WI-FI router to the operating station, where they are subjected to a detailed computer-based analysis. From the identification and deeper analysis of the measured signals it is possible to monitor the optimum operating conditions of the vibration equipment, depending on predetermined parameters, namely, the trough inclination angle, the throw angle, the rotor speed of the vibration motor, the spring stiffness and the amount of material on the trough surface. The highest mean magnitude of the effective vibration velocity (4.8 mm·s^−1^) in the vertical direction was measured on a model vibrating conveyor, with rubber springs with a stiffness of 54 N·mm^−1^, with the unloaded trough without the conveyed material. The lowest mean magnitude of the effective vibration velocity was 1.2 mm·s^−1^ in the vertical direction with a weight of 5.099 kg of conveyed material on the trough. Suitably designed rubber springs, of optimum stiffness, dampen the vibrations transmitted to the machine frame. From their sizes, it is possible to remotely monitor the working operation of the vibrating conveyor or to obtain information about the failure of one or several used rubber springs.

## 1. Introduction

In vibrating conveyors or separators the vibration is considered to be the movement of the trough and other structural parts, individual parts that oscillate around the equilibrium position [1].

The level of vibration of the trough transmitted to the frame or foundation, to which the frame of the vibrating machines [2] is mechanically attached, is significantly influenced by a number of parameters, the most important of which are the weight of the trough not filled with material and the rotor speed of the vibrating motor.

In condition monitoring, vibration measurement can be used to indicate and monitor the operational performance of vibrating conveyors or vibrating sorters. It is the aim of this paper to demonstrate that from the analysis of the measured vibration signals on the frame of the vibrating conveyor using sensors, it is possible to remotely monitor its working activity and obtain information about the mass or volume quantity [3] of the conveyed or sorted material located on the trough.

Vibrations of the trough or frame of the vibrating conveyor [4] are given by means of a combination of six movements—displacement in the orthogonal system of x, y, z coordinates and rotation around these x, y, z axes (such a mechanical system has six degrees of freedom). However, the majority of mechanical systems do not oscillate as a single fixed point; waves are generated instead [5].

Harmonic vibrations are periodic vibrations (i.e., the time course of the vibrodiagnostic quantities is repeated) containing a single frequency. The harmonic vibration is beneficial in that, if we determine at least one of the fundamental quantities (deflection, velocity, or acceleration), then the remaining quantities can be calculated according to the relations [6,7].

In [8], P. Czubak and M. Gajowy attempt to identify important elements determining the design of an anti-resonant-type conveyor. Their article analyses the impacts of various types of suspension and the ratio of the body mass to the mass of the conveyor trough on forces transmitted to the ground.

M. Sturm, in [9], presents a dynamic model, based on a two-mass absorber system, helping to avoid the transfer of vibrations to the ground.

In [10], G. Cieplok presents a spatial model of a vibrating conveyor supported on steel-elastomer vibration isolators and vibrated by two inertial vibrators. Results of analyses of the effect of the layout of vibrators on the conveyor operation were attained.

In [11], a new vibrating conveyor designed for precise material dosing was investigated. The transport possibilities in the circum-resonance zone were tested analytically as well as by simulation. The optimal working point of the system, allowing the vibration amplitude of the eliminator to be lowered on its own suspension, was found.

M. Sturm, in [12] established a dynamic model for the targeted displacement of the centre of elasticity of a one-mass conveyor, enabling an optimal motion of the conveyor by an optimized set of springs, connecting the conveying element to the frame.

In [13], C.Y. Lee et al. propose an innovative design that uses a composite sinusoidal control signal to drive a linear two-stage piezoelectric vibratory feeder. The dynamic characteristics of this feeder were both investigated by theoretical formulation and experimental measurement. It is found that the conveying performance of this feeder is better suited for using the composite signal with two sinusoidal components of double frequency.

The mathematical model of the single-mass vibratory screening conveyor equipped with the proposed exciter is developed using the Lagrange-d’Alembert principle, see V. Korendiy et al. [14]. The simulation of the system’s kinematic and dynamic characteristics under different operational conditions is carried out in the Mathematica software using the Runge-Kutta numerical integrated methods.

In general, the motion of bodies in mechanics (including the vibration of vibrating conveyor troughs or vibrating screen surfaces) can be described by the acceleration a(t) [m·s^−2^], velocity v(t) [m·s^−1^], displacement y(t) [m], frequency f [Hz] or period T = f^−1^ [s^−1^]. All these variables are mathematically interrelated.

The oscillation rate is used at low and medium frequencies (f = 10 Hz to 1 kHz) to identify disturbances manifested in these frequencies. Consequently, the vibration rate time history is used for the analysis at lower machine revolutions (for higher frequencies of f = 1 kHz to 20 kHz the vibration acceleration is used).

In [15], W. Zmuda and P. Czubak determined the dependence of the transport speed of the tested conveyor as a function of the excitation frequency. Favourable excitation frequencies at transports in the main and reversal directions were found, and the high usefulness of the machine in the production lines requiring accurate material dosage was indicated.

The amplitude of the harmonic oscillation (A = _ymax_(t) [m] − maximum deflection) is often replaced by another characteristic to describe the harmonic signal: mean value = 0.637-amplitude of oscillation (y_ave_ = 0.637∙A [m]) or RMS value (RMS) = 0.707∙amplitude of oscillation (y_RMS_ = 0.707∙A [m] [6]). The effective vibration value y_RMS_ [m] (the so-called root mean square value), replaces the vibration time history y(t) [m] with a constant energy value.

A theoretical analysis of the synchronization of inertial vibrators of a vibratory conveyor with a dynamic damper is presented in [16]. This paper builds on the findings of papers [10,14] and presents a more detailed analysis of synchronization processes in anti-resonant devices. While [14] focuses on asymmetric vibrators and their effect on the dynamic behaviour of the device, ref. [16] deepens this approach with an emphasis on optimizing the synchronization of drive vibrators. Compared to [10], which focuses on self-synchronization in conventional vibratory conveyors, this paper highlights the unique characteristics of anti-resonance systems.

The paper [17] is about the implementation of a two-stage vibration insulation system for a “Shake-Out” conveyor. Its authors, W. Fiebiech and J. Wrobel, describe the effect of various insulation levels on reducing the transmission of vibrations to the equipment frame and surrounding structures.

The article [18] presents a method of vibrating screen trajectory control based on MR (magnetorheological) dampers applied in a screen suspension. Authors A. Ogonowski and P. Krauze present a mathematical description of the dynamic screen model was derived, and the parameters of this model were estimated based on experimental data from a semi-industrial vibrating screen.

The new type of reversible, vibratory conveyor in which a smooth velocity control (within the full range) in both directions is possible, is presented in [19].

The paper [20] focuses on the issue of vibration insulation of vibrating conveyors and its effect on the reduction in vibration transmission to the structural frame and foundations. The authors analyse various types of vibration insulation elements, such as rubber dampers, spring systems and viscous dampers, and evaluate their effectiveness under various operating conditions. This study encompasses both theoretical analysis and experimental measurements that confirm the effectiveness of various insulation methods. In contrast to [10], which deals with the general effects of physical parameters on vibration insulation, this paper focuses more on specific insulation methods and their practical application. Compared to [17], which describes two-stage vibration insulation, this text provides a more in-depth analysis of various insulation technologies and their effectiveness at various vibration frequencies.

The new type of reversible, vibratory conveyor in which a smooth velocity control (within the full range) in both directions is possible, is presented in [21,22]. The study [21] combines mathematical modelling and laboratory tests to investigate the influence of system parameters on the synchronization effect.

The paper [23] is devoted to the analysis of operational properties of vibratory conveyors, whose principle of operations is based on Frahm’s dynamic elimination effect. In order to investigate the correctness of the machine’s operation, the physical model of the conveyor with a feed [24] was constructed, and then its mathematical model was developed. The paper [24] is complementary to [20], which deals with vibration insulation, but instead of rubber and air springs, leaf springs are examined.

The paper [25] deals with modelling and experimental verification of the dynamic behaviour of a vibrating conveyor with a controlled centrifugal exciter. This study investigates the effectiveness of vibration transmission and how to optimise its settings. This paper extends the findings of [14] with experimental investigations and adds a more detailed analysis of controlled vibration in material sorting applications.

In [26], P. Czubak and W. Surowka present the design and experimental testing of a self-adjusting vibrating conveyor that automatically adapts to changes in load and operating conditions. The authors analys its performance characteristics and the possibilities of its deployment in industry. This paper is complementary to [19], which focuses on the controlled change in of speed and direction of the conveyor’s movement. However, in this case, the authors investigate the automatic adaptability of the conveyor to various operating conditions.

## 2. Materials and Methods

The experimental device, (see Figure 1), has been designed at the Department of Machine and Industrial Design, Faculty of Mechanical Engineering, VSB-Technical University of Ostrava, in the software environment of SolidWorks^®^ Premium 2012 × 64 SP5.0.

The experimental device consists of frame 1 (made of AL profiles of cross-section 20 × 20 mm, type MI 20 × 20 I5 [27]), trough 2 (made by “Valter Špalek—Plexi” from Plexiglas 6 mm thick) and supporting elements including rubber springs 3. The vibrating motor 4 [28] (power P_e_ = 40 W, speed n_e_ = 3000·min^−1^) is attached to trough 2, which can be tilted by an angle of β = 0 to 15 deg with respect to the horizontal plane, by means of screw connections.

On the rotor shaft of the asynchronous single-phase vibration motor 4 there are two eccentric weights, each with a mass m_o_ [kg] (m_o_ = 80.44 · 10^−3^ kg; see 3D model, Figure 2c, created in SolidWorks^®^ Premium 2012 × 64 SP5.0 [29]). Eccentricity e_u_ [m], (see Figure 2b), expresses the radius of rotation of the centre of gravity T of the weight.(1)$$F_{c1}=m_{0}\cdot e_{u}\cdot \omega_{a}^{2}\left[N\right];\;F_{c}=2\cdot F_{c1}=2\cdot m_{0}\cdot e_{u}\cdot \omega_{a}^{2}=2\cdot m_{0}\cdot \frac{v_{a}^{2}}{e_{u}}\left[N\right],$$

When the motor rotor rotates at the actual speed n_ea_ [s^−1^], each eccentric weight (see Figure 2c) of the vibration motor generates the centrifugal force F_c1_ [N] (1), assuming that ω_a_ [rad·s^−1^] (2) is the instantaneous angular velocity of the rotating weight.(2)$$\omega_{a}=2\cdot \pi \cdot f=2\cdot \pi \cdot \frac{1}{T}=2\cdot \pi \cdot n_{ea}\left[rad\cdot s^{-1}\right]\to f=n_{ea}\left[s^{-1}\right],$$

Trough oscillation frequency f [Hz], when using electromagnetic exciters [30], is usually identical to the AC frequency of 50 Hz in Europe (60 Hz in the U.S.); it represents the number of cycles of the T sinusoid per second.

The instantaneous value of the centrifugal force F_c_ [N] in the axis of oscillation (which is inclined at an angle of α = 10° to 25° to the horizontal plane, the so-called throw angle α [deg]; (see Figure 1b) can be quantified according to (1).

The force F_c_ [N] (1) must be maintained in a continuous oscillating motion with predetermined oscillation parameters (deflection, frequency f [Hz], angle of throw α [deg]) the mass m_t_ [kg], which is given by the sum of masses of the trough of the vibrating conveyor, the exciter and the material conveyed on the trough.

The trough of the vibrating conveyor 2, to which the vibrating motor 4 is mechanically attached at an angle of α [deg], oscillates with harmonic (periodic) motion. The instantaneous deflection y(t) [m] (3) of the periodically oscillating trough (it takes both positive and negative values) expresses the instantaneous distance of the centre of gravity T of trough 2 from its equilibrium position.(3)$$y\left(t\right)=e_{u}\cdot \sin\left(\varphi \right)=e_{u}\cdot \sin\left(\omega_{a}\cdot t\right)=e_{u}\cdot \sin\left(2\cdot \pi \cdot n_{ea}\cdot t\right)\left[m\right],$$

The diagram expressing the dependence of the instantaneous deflection y(t) [m] on time t [s] is called a time diagram and has the shape of a sinusoid, see Figure 3a.

A condition for the transport of material by a vibratory conveyor with a microblast is that the highest value of the acceleration a_2_ [m·s^−2^] (i.e., vhen ω·t = −1) acting on the material grain, (see Figure 3b), is higher than the gravitational acceleration g [m·s^−2^].

The angular frequency f [s^−1^] is defined as the inverse value of the period T, i.e., the frequency (see Figure 3), namely the mean number of revolutions of n_ea_ [s^−1^] per unit time. In the case of uniform motion along a circle, the angular frequency f [s^−1^] is numerically equal (if the angular velocity is considered as a scalar) to the angular velocity ω_a_ [rad·s^−1^] (2).

The trough oscillates at a frequency f [Hz] at an angle of α [deg] to the direction of traffic (to the horizontal plane). The bottom of the trough, on which the material to be conveyed rests, is alternately in the upper and lower extreme positions.

The component of oscillation in the direction y_x_(t) [m], (see Figure 4a), and the component in the direction perpendicular to y_y_(t) [m] to the direction of material motion is described by Equation (4).(4)$$y_{x}\left(t\right)=e_{u}\cdot \cos\left(\alpha \right)\cdot \sin\left(\omega_{a}\cdot t\right)[m];y_{y}\left(t\right)=e_{u}\cdot \sin\left(\alpha \right)\cdot \sin\left(\omega_{a}\cdot t\right)[m],$$

The instantaneous deflection y(t) [m] of the harmonic vibration of the trough described by Equation (3), and the components of the vibration (assuming trough inclination angle β = 0 deg) in the direction y_x_(t) [m] (4) and in the direction perpendicular y_y_(t) [m] (4) to the direction of material movement, with amplitude e_u_ [m] (eccentricity of unbalance e_u_ = 12.95 mm) is presented in Figure 4b. If the trough is inclined at an angle β [deg] with respect to the horizontal plane, then the oscillation components can be expressed by Equation (5).(5)$$y_{x}^{\prime }\left(t\right)=\frac{y_{x}\left(t\right)}{\cos\left(\beta \right)}=\frac{e_{u}\cdot \cos\left(\alpha \right)\cdot \sin\left(\omega_{a}\cdot t\right)}{\cos\left(\beta \right)}\left[m\right];y_{y}^{\prime }\left(t\right)=\frac{y_{y}\left(t\right)}{\cos\left(\beta \right)}=\frac{e_{u}\cdot \sin\left(\alpha \right)\cdot \sin\left(\omega_{a}\cdot t\right)}{\cos\left(\beta \right)}[m],$$

The harmonic oscillation of the trough mounted on springs of total stiffness s_s_ [N·mm^−1^] is caused by the force F_a_ [N] (6), the magnitude of which is directly proportional to the deflection y(t) [m] and has a direction to the equilibrium position at each moment.(6)$$F_{a}=s_{s}\cdot y\left(t\right)=s_{s}\cdot e_{u}\cdot \sin\left(\omega_{a}\cdot t\right)\left[N\right],$$

The spring stiffness (constant of proportionality) s_s_ [N·m^−1^] [31] is defined as the fraction of the force F [N] that elongates/shortens the spring by the value ΔL [m].

The mass of the oscillating system m_t_ [kg] can be determined according to relation (7), where m_e_ [kg] is the mass of the vibrating motor (m_e_ = 1.372 kg, see [28]); m_z_ [kg] is the mass of the plastic trough (including connecting material) (m_z_ = 1.653 kg, determined by weighing); m_m_ [kg] is the mass of the material on the trough (0 kg; 2.570 kg and 5.099 kg).(7)$$m_{t}=m_{e}+m_{z}+m_{m}[kg],$$

The natural frequency of the oscillating system f_0_ [s^−1^] determined from the angular velocity ω_0_ [rad·s^−1^] of the mass m_t_ [kg] can be expressed according to relation (8).(8)$$\omega_{0}=\sqrt{\frac{s_{s}}{m_{t}}}\left[rad\cdot s^{-1}\right]\;\to f_{0}=\frac{1}{2\cdot \pi }\cdot \sqrt{\frac{s_{s}}{m_{t}}}\left[s^{-1}\right]$$

From the ratio of the working frequency of the oscillating system f [s^−1^] (2) and the natural frequency of the oscillating system f_0_ [s^−1^] (7), it is possible to determine, in which region the vibrating machine will work (z = f_a_/f_0_ [-]), if z < 1—is a sub-resonant region, z = 0.85 ÷ 0.95 is a—resonant region, z > 1 ÷ 5 is a—supra-resonant region.

When a trough assembly with mass m_t_ = 8.124 kg is placed on 4 rubber springs (lengths in the free state: L_0_ = 20 mm, D = 18 mm, and s_s_ = 54 N·mm^−1^ [32]), a rubber spring with size ΔL_1_ = 0.37 mm will be compressed due to the applied weight G_t_ = m_t_·g = 79.76 N, see Figure 5. Due to the centrifugal force F_c_ = 15.15 N (1) generated by the eccentric weights of the vibration motor, at the speed n_ea_ = 787 min^−1^, one spring of ΔL_2_ = 0.07 mm is compressed. According to the autors of [7] the maximum permissible compression of the rubber spring is 5.0 mm, which is a greater value than ΔL_c_ = ΔL_1_ + ΔL_2_ = 0.44 mm.

The magnitude of the centrifugal force F_c_ [N] (1) generated by the eccentric weights (mass m_o_ = 80.44·10^−3^ kg) of the vibration motor reaches the maximum (F_c(max)_ = 205.62 N) at a moment when the rotor of the vibration motor is rotating at speed n_e_ = 3000 min^−1^. The maximum allowable compression of the rubber spring is ΔL_c(max)_ = 1.32 mm, which is less than the maximum allowable compression (5.0 mm [33]) of the rubber spring specified by the [34].

Speed of asynchronous single-phase motor 1 of vibration exciter, see Figure 6, was controlled by a thyristor electronic voltage regulator 2 typ DJ-SC40 [14].

The actual revolutions of the electric motor 1 n_ea_ [min^−1^] were obtained by measuring the speed sensor UNI-T UT373 3 [35]. The rotor revolutions of the asynchronous single-phase motor were monitored with laboratory measurements by an optical laser sensor DS-TACHO-3 4 [36]. Compressing 1 piece of rubber spring by ΔL_2_ [m], (see Figure 5), directly proportionally affects the revolutions n_ea_ [s^−1^] of the vibration motor. At maximum speed n_e_ = 3000·min^−1^ of the rotor of the vibration motor, the centrifugal speed would be (1) F_c_ = 205.62 N.

Vibrations generated by rotating (n_ea_ [s^−1^] revolutions) eccentric weights (see Figure 2) of the vibration motor, transmitted to the frame of the vibration conveyor, were detected by two acceleration sensors 5 (type PCE KS903.10) [37].

The signals from the acceleration sensors were recorded during the experimental measurements with the measuring apparatus Dewesoft SIRIUSi-HS 6xACC, 2xACC+ [38]; see Figure 7. The records of the measured values were transformed by the measuring apparatus to the effective values of the broadband velocity [mm·s^−1^] in the range of 10 ÷ 1 · 10^4^ Hz (this frequency range is applied to the ISO 10816-3 [39]). The effective values of the vibration velocity i_RMS(α,β,m,ne)_ [mm·s^−1^] (where i = x, y, z—spatial orientation of the axes of the coordinate system) of the periodic waveform were displayed on the PS monitor in the environment of the measuring software Dewesoft X (version number 2024.5, release-241202, 64-bit) [40].

## 3. Results

Three repeated measurements of the effective vibration velocity i_RMS(α,β,m,ne)_ [mm·s^−1^] [6,7] of the Plexiglas trough and the support frame of the laboratory equipment were carried out under the same technical conditions for various input parameters, which are as follows: trough inclination angle β = 0, 5, 10 and 15 deg; throw angle α = 10 to 25 deg; load mass m_m_ = 0 kg, 2.57 kg, and 5.099 kg; actual speed n_ea_ [s^−1^] of the rotor of the vibrating electric motor. The trough of the vibrating conveyor model is supported by four pieces of rubber springs. In Section 3.1, Section 3.2, Section 3.3 and Section 3.4, rubber springs (silent blocks) made of natural rubber (NR), with a diameter of 18 mm, unloaded length of H_0_ = 20 mm, hardness of 45° Shore, and stiffness s_s_ = 54 N·mm^−1^, are used.

One of the acceleration sensors is attached to the upper surface of the Plexiglas trough before measurement, while another acceleration sensor is attached to the horizontal surface of the AL profile of the vibrating conveyor frame. A weight of mass m_m_ [kg] is placed on the trough of the vibrating conveyor, inclined at an angle of β [deg]. The electronic speed controller sets the desired revolutions n_ea_ [min^−1^] of the rotor of the vibration motor. For different input parameters α [deg], β [deg], m_m_ [kg], and n_ea_ [min^−1^], vibrations are sensed, and the measured signals are converted to effective vibration rate values, i_RMS(α,β,m,ne)_ [mm·s^−1^], which are entered in the tables below. Since the vibration motor is attached to the trough by bolted connections, the trough’s angle of throw is reduced to α = 25 deg − β [deg] when the trough is deflected (i.e., when the angle of inclination β [deg] is changed) from the parallel position.

Figure 8 presents a randomly selected record of vibration measurements by acceleration sensors (type PCE KS903.10) attached to the upper surface of the trough (diagram in the upper left part of the figure) and to the frame (diagram in the lower left part of the figure) of the vibrating conveyor model in the DEWESoftX software environment. Records of measured effective velocity values i_RMS(α,β,m,ne)_ [mm·s^−1^] in Figure 8 (and given in the tables in chapters 3.1 to 3.4) present the measured vibration values in the i = x, y, z coordinate axes (x-axis—blue, y-axis—red, and z-axis—green).

In the tables, (see Section 3.1, Section 3.2, Section 3.3, Section 3.4, Section 3.5, Section 3.6, Section 3.7 and Section 3.8), giving the measured effective vibration velocity values i_RMS(α,β,m,ne)_ [mm·s^−1^] and the values of i_RMS(α,β,m,ne)A_ [mm·s^−1^], which is the arithmetic mean of all (n = 3; the number of repeated measurements) measured vibration values of i_RMS(α,β,m,ne)_ [mm·s^−1^] and κ_α,n_ [mm·s^−1^], which is the marginal error.—t_α,n_ [-] (t_5%,3_ = 4.3) is the Student coefficient for the risk α [%] (α = 5%) and the confidence coefficient P [%] (P = 95%) [41].

### 3.1. Trough Inclination Angle β = 0 deg, Throw Angle α = 25 deg, and Silent Block 18 × 20 mm M6x10

Table 1 lists the measured effective vibration velocity values of i_RMS(α,β,m,ne)_ [mm·s^−1^] sensed on the surface of the Plexiglas trough and on the supporting structure of the vibrating conveyor for the input values α = 25 deg, β = 0 deg, m_m_ = 0 kg, and n_ea_ = 833 min^−1^.

Figure 9a,b present the measured effective vibration velocity values i_RMS(α,β,m,ne)_ [mm·s^−1^] detected by the PCE KS903.10 acceleration sensors on the trough surface (unladen with material weight) and on the vibrating conveyor frame, at a vibrating motor rotor revolutions value of 13.88 s^−1^.

Figure 9c,d presents the measured effective vibration velocity values i_RMS(α,β,m,ne)_ [mm·s^−1^] detected by the PCE KS903.10 acceleration sensors on the trough surface (loaded with material of mass m_m_ = 2.57 kg) and on the vibrating conveyor frame, at a vibrating motor rotor revolutions of 13.83 s^−1^.

Table 2 lists the measured effective vibration velocity values of i_RMS(α,β,m,ne)_ [mm·s^−1^], in three planes perpendicular to each other, sensed on the surface of the Plexiglas trough and on the frame of the vibrating conveyor for input values α = 25 deg, β = 0 deg, m_m_ = 2.57 kg, and n_ea_ = 830 min^−1^.

Table 3 lists the measured effective vibration velocity values of i_RMS(α,β,m,ne)_ [mm·s^−1^], in three planes perpendicular to each other, sensed on the surface of the Plexiglas trough and on the frame of the vibrating conveyor for input values α = 25 deg, β = 0 deg, m_m_ = 5.099 kg, and n_ea_ = 839 min^−1^.

Figure 10a,b present the measured effective vibration velocity values i_RMS(α,β,m,ne)_ [mm·s^−1^] detected by the PCE KS903.10 acceleration sensors on the trough surface (loaded with material of mass m_m_ = 5.099 kg) and on the vibrating conveyor frame, at a vibrating motor rotor revolutions value of 13.98 s^−1^.

Figure 9 and Figure 10 present the time histories of the measured effective vibration velocity values (in three mutually perpendicular x, y, and z axes), which were randomly selected from the three repeated measurements reported in Table 1 through Table 4.

### 3.2. Trough Inclination Angle β = 5 deg, Throw Angle α = 20 deg, and Silent Block 18 × 20 mm M6x10

Table 4 lists the measured effective vibration velocity values of i_RMS(α,β,m,ne)_ [mm·s^−1^], in three planes perpendicular to each other, sensed on the surface of the Plexiglas trough and on the frame of the vibrating conveyor for input values α = 20 deg, β = 5 deg, m_m_ = 0 kg, and n_ea_ = 806 min^−1^.

Figure 10c,d present the measured effective vibration velocity values i_RMS(α,β,m,ne)_ [mm·s^−1^] detected by the PCE KS903.10 acceleration sensors on the trough surface (unladen with material weight) and on the vibrating conveyor frame, at a vibrating motor rotor revolution value of 13.43 s^−1^.

Table 5 lists the measured effective vibration velocity values of i_RMS(α,β,m,ne)_ [mm·s^−1^], in three planes perpendicular to each other, sensed on the surface of the Plexiglas trough and on the frame of the vibrating conveyor for input values α = 20 deg, β = 5 deg, m_m_ = 2.57 kg. and n_ea_ = 862 min^−1^.

Table 6 lists the measured effective vibration velocity values of i_RMS(α,β,m,ne)_ [mm·s^−1^], in three planes perpendicular to each other, sensed on the surface of the Plexiglas trough and on the frame of the vibrating conveyor for input values α = 20 deg, β = 5 deg, m_m_ = 5.099 kg, and n_ea_ = 879 min^−1^.

### 3.3. Trough Inclination Angle β = 10 deg, Throw Angle α = 15 deg, and Silent Block 18 × 20 mm M6x10

Table 7 lists the measured effective vibration velocity values of i_RMS(α,β,m,ne)_ [mm·s^−1^], in three planes perpendicular to each other, sensed on the surface of the Plexiglas trough and on the frame of the vibrating conveyor for input values α = 15 deg, β = 10 deg, m_m_ = 0 kg, and n_ea_ = 816 min^−1^.

Table 8 lists the measured effective vibration velocity values of i_RMS(α,β,m,ne)_ [mm·s^−1^], in three planes perpendicular to each other, sensed on the surface of the Plexiglas trough and on the frame of the vibrating conveyor for input values α = 5 deg, β = 10 deg, m_m_ = 2.57 kg, and n_ea_ = 961 min^−1^.

Table 9 lists the measured effective vibration velocity values of i_RMS(α,β,m,ne)_ [mm·s^−1^], in three planes perpendicular to each other, sensed on the surface of the Plexiglas trough and on the frame of the vibrating conveyor for input values α = 15 deg, β = 10 deg, m_m_ = 5.099 kg, and n_ea_ = 917 min^−1^.

### 3.4. Trough Inclination Angle β = 15 deg, Throw Angle α = 10 deg, and Silent Block 18 × 20 mm M6x10

Table 10 lists the measured effective vibration velocity values of i_RMS(α,β,m,ne)_ [mm·s^−1^], in three planes perpendicular to each other, sensed on the surface of the Plexiglas trough and on the frame of the vibrating conveyor for input values α = 10 deg, β = 15 deg, m_m_ = 0 kg, and n_ea_ = 748 min^−1^.

Table 11 lists the measured effective vibration velocity values of i_RMS(α,β,m,ne)_ [mm·s^−1^], in three planes perpendicular to each other, sensed on the surface of the Plexiglas trough and on the frame of the vibrating conveyor for input values α = 10 deg, β = 15 deg, m_m_ = 2.57 kg, and n_ea_ = 853 min^−1^.

Figure 11a,b present the measured effective vibration velocity values i_RMS(α,β,m,ne)_ [mm·s^−1^] detected by the PCE KS903.10 acceleration sensors on the trough surface (loaded with material of mass m_m_ = 2.57 kg) and on the vibrating conveyor frame, at a vibrating motor rotor revolutions value of 15.88 s^−1^.

Figure 11c,d present the measured effective vibration velocity values i_RMS(α,β,m,ne)_ [mm·s^−1^] detected by the PCE KS903.10 acceleration sensors on the trough surface (loaded with material of mass m_m_ = 5.099 kg) and on the vibrating conveyor frame, at a vibrating motor rotor revolutions value of 15.80 s^−1^.

Table 12 lists the measured effective vibration velocity values of i_RMS(α,β,m,ne)_ [mm·s^−1^], in three planes perpendicular to each other, sensed on the surface of the Plexiglas trough and on the frame of the vibrating conveyor for input values α = 10 deg, β = 15 deg, m_m_ = 5.099 kg, and n_ea_ = 948 min^−1^.

In the above tables, the measured effective vibration velocity values i_RMS(α,β,m,ne)_ [mm·s^−1^] are affected by the actual magnitude of the centrifugal force F_c1_ [N] (1). The size of F_c1_ [N] determines the actual rotor revolutions n_ea_ [s^−1^] of the vibration motor. The centrifugal force F_c1_ [N], (see Figure 12a), increases quadratically with increasing angular velocity ω_a_ [rad·s^−1^].

Figure 12b gives the mean values of the three times the effective velocity values were measured in the vertical direction (see Table 1, Table 2, Table 3, Table 4, Table 5, Table 6, Table 7, Table 8, Table 9, Table 10, Table 11 and Table 12) and, measured on the frame of the vibrating conveyor model. The curves (depicted in colour at in Figure 12b), which are formed by two lines (passing through three points), are described by the powers of the trend lines (which are the functions that best describe the measured data).

In order to compare the measured vibration values i_RMS(α,β,m,ne)_ [mm·s^−1^] (see Table 1, Table 2, Table 3, Table 4, Table 5, Table 6, Table 7, Table 8, Table 9, Table 10, Table 11 and Table 12, Section 3.1, Section 3.2, Section 3.3 and Section 3.4) measurements were made on the trough surface and on the frame of the vibration machine model, a plastic trough supported on four rubber springs with an 18 mm diameter, a length in the unloaded state of H_0_ = 20 mm, using rubber springs with a 25 mm diameter, and a length in the unloaded state of H_0_ = 15 mm.

Section 3.5, Section 3.6, Section 3.7 and Section 3.8 indicate the measured effective vibration velocity values i_RMS(α,β,m,ne)_ [mm·s^−1^] in three axes perpendicular to each other, for various input parameters, which are as follows: trough inclination angle β = 0, 5, 10, the 15 deg; throw angle α = 0 to 25 deg; load mass m_m_ = 0 kg, 2.570 kg and 5.099 kg; and actual revolutions n_ea_ [s^−1^] of the rotor of the vibrating electric motor. Rubber springs (silent blocks) made of natural rubber, with a diameter of 25 mm, unloaded length of H_0_ = 15, stiffness of s_s_ = 120 N·mm^−1^, and a hardness of 45° Shore, were used. The maximum permissible compression of the rubber spring is according to [33], 3.75 mm.

When a trough assembly with mass m_t_ = 8.124 kg is placed on four rubber springs (lengths in the free state L_0_ = 15 mm, D = 25 mm, and s_s_ = 120 N·mm^−1^), a rubber spring with size ΔL_1_ = 0.17 mm will be compressed due to the applied weight G_t_ = m_t_·g = 79.76 N, see Figure 5. Due to the centrifugal force F_c_ = 16.51 N (1) generated by the eccentric weights of the vibration motor, at the speed n_ea_ = 850 min^−1^, one spring of ΔL_2_ = 0.03 mm is compressed. According to [33] the maximum permissible compression of the rubber spring is 3.75 mm, which is greater value than ΔL_c_ = ΔL_1_ + ΔL_2_ = 0.2 mm. The magnitude of the centrifugal force F_c_ [N] (1) generated by the eccentric weights (m_o_ = 80.44·10^−3^ kg) of the vibration motor reaches the maximum (F_c(max)_ = 205.62 N) at a moment when the rotor of the vibration motor is rotating at a speed of n_e_ = 3000 min^−1^. The maximum allowable compression of the rubber spring is ΔL_c(max)_ = 0.59 mm, which is less than the maximum allowable compression (3.75 mm) of the rubber spring specified by the manufacturer.

Not all measured effective vibration velocity values (in three mutually perpendicular x, y, and z axes), for three repeated measurements under the same technical conditions, indicated in the tables below in Section 3.5, Section 3.6, Section 3.7 and Section 3.8 (see Table 13), will be presented due to the scope limitations of this paper—not even randomly, selected graphical records of time records of measured effective vibration velocity values (as presented in Figure 13 and Figure 14) by the measuring apparatus Dewesoft SIRIUSi-HS 6xACC, 2xACC+ [38].

### 3.5. Trough Inclination Angle β = 0 deg, Throw Angle α = 25 deg, and Silent Block 25 × 15 mm M6x12

Three repeated measurements of the effective vibration velocity i_RMS(α,β,m,ne)_ [mm·s^−1^] [6,7] of the Plexiglas trough and the support frame of the laboratory equipment were carried out under the same technical conditions for various input parameters, which are as follow: trough inclination angle β = 0, 5, 10 and 15 deg; throw angle α = 10 to 25 deg; load mass m_m_ = 0 kg, 2.57 kg and 5.099 kg; the actual speed n_ea_ [s^−1^] of the rotor of the vibrating electric motor. The trough of the vibrating conveyor model is supported by four pieces of rubber springs. In Section 3.5, Section 3.6, Section 3.7 and Section 3.8, rubber springs (silent blocks) made of natural rubber (NR) with a diameter of 25 mm, unloaded length of H_0_ = 15 mm, hardness of 45° Shore, and stiffness of s_s_ = 1834 N·mm^−1^ [32,33] are used.

Table 13 lists the measured effective vibration velocity values of i_RMS(α,β,m,ne)_ [mm·s^−1^], in three planes perpendicular to each other, sensed on the surface of the Plexiglas trough and on the frame of the vibrating conveyor for input values α = 25 deg, β = 0 deg, m_m_ = 0 kg a n_ea_ = 761 min^−1^.

Table 14 lists the measured effective vibration velocity values of i_RMS(α,β,m,ne)_ [mm·s^−1^], in three planes perpendicular to each other, sensed on the surface of the Plexiglas trough and on the frame of the vibrating conveyor for input values α = 25 deg, β = 0 deg, m_m_ = 2.57 kg, and n_ea_ = 830 min^−1^.

Table 15 lists the measured effective vibration velocity values of i_RMS(α,β,m,ne)_ [mm·s^−1^], in three planes perpendicular to each other, sensed on the surface of the Plexiglas trough and on the frame of the vibrating conveyor for input values α = 25 deg, β = 0 deg, m_m_ = 5.099 kg, and n_ea_ = 750 min^−1^.

### 3.6. Trough Inclination Angle β = 5 deg, Throw Angle α = 20 deg, and Silent Block 25 × 15 mm M6x12

Table 16 lists the measured effective vibration velocity values of i_RMS(α,β,m,ne)_ [mm·s^−1^], in three planes perpendicular to each other, sensed on the surface of the Plexiglas trough and on the frame of the vibrating conveyor for input values α = 20 deg, β = 5 deg, m_m_ = 0 kg, and n_ea_ = 847 min^−1^.

Table 17 lists the measured effective vibration velocity values of i_RMS(α,β,m,ne)_ [mm·s^−1^] sensed on the surface of the Plexiglas trough and on the supporting structure of the vibrating conveyor for the input values α = 20 deg, β = 5 deg, m_m_ = 2.57 kg, and n_ea_ = 828 min^−1^.

Figure 13a,b present the measured effective vibration velocity values i_RMS(α,β,m,ne)_ [mm·s^−1^] detected by the PCE KS903.10 acceleration sensors on the trough surface (loaded with material of mass m_m_ = 2.57 kg) and on the vibrating conveyor frame, at a vibrating motor rotor revolutions value of 12.08 s^−1^.

Table 18 lists the measured effective vibration velocity values of i_RMS(α,β,m,ne)_ [mm·s^−1^], in three planes perpendicular to each other, sensed on the surface of the Plexiglas trough and on the frame of the vibrating conveyor for input values α = 20 deg, β = 5 deg, m_m_ = 5.099 kg, and n_ea_ = 746 min^−1^.

Figure 13c,d present the measured effective vibration velocity values i_RMS(α,β,m,ne)_ [mm·s^−1^] detected by the PCE KS903.10 acceleration sensors on the trough surface (loaded with material of mass m_m_ = 5.099 kg) and on the vibrating conveyor frame, at a vibrating motor rotor revolution value of 12.43 s^−1^.

### 3.7. Trough Inclination Angle β = 10 deg, Throw Angle α = 15 deg, and Silent Block 25 × 15 mm M6x12

Table 19 lists the measured effective vibration velocity values of i_RMS(α,β,m,ne)_ [mm·s^−1^] sensed on the surface of the Plexiglas trough and on the supporting structure of the vibrating conveyor for the input values α = 15 deg, β = 10 deg, m_m_ = 0 kg, and n_ea_ = 817 min^−1^.

Figure 14a,b present the measured effective vibration velocity values i_RMS(α,β,m,ne)_ [mm·s^−1^] detected by the PCE KS903.10 acceleration sensors on the trough surface (unladen with material weight) and on the vibrating conveyor frame, at a vibrating motor rotor revolution value of 13.62 s^−1^.

Table 20 lists the measured effective vibration velocity values of i_RMS(α,β,m,ne)_ [mm·s^−1^], in three planes perpendicular to each other, sensed on the surface of the Plexiglas trough and on the frame of the vibrating conveyor for input values α = 15 deg, β = 10 deg, m_m_ = 2.57 kg, and n_ea_ = 835 min^−1^.

Figure 14c,d present the measured effective vibration velocity values i_RMS(α,β,m,ne)_ [mm·s^−1^] detected by the PCE KS903.10 acceleration sensors on the trough surface (loaded with material of mass m_m_ = 2.57 kg) and on the vibrating conveyor frame, at a vibrating motor rotor revolution value of 13.92 s^−1^.

Table 21 lists the measured effective vibration velocity values of i_RMS(α,β,m,ne)_ [mm·s^−1^] sensed on the surface of the Plexiglas trough and on the supporting structure of the vibrating conveyor for the input values α = 15 deg, β = 10 deg, m_m_ = 5.099 kg, and n_ea_ = 779 min^−1^.

### 3.8. Trough Inclination Angle β = 15 deg, Throw Angle α = 10 deg, and Silent Block 25 × 15 mm M6x12

Table 22 lists the measured effective vibration velocity values of i_RMS(α,β,m,ne)_ [mm·s^−1^] sensed on the surface of the Plexiglas trough and on the supporting structure of the vibrating conveyor for the input values α = 10 deg, β = 15 deg, m_m_ = 0 kg, and n_ea_ = 849 min^−1^.

Table 23 lists the measured effective vibration velocity values of i_RMS(α,β,m,ne)_ [mm·s^−1^], in three planes perpendicular to each other, sensed on the surface of the Plexiglas trough and on the frame of the vibrating conveyor for input values α = 10 deg, β = 15 deg, m_m_ = 2.57 kg, and n_ea_ = 751 min^−1^.

Table 24 lists the measured effective vibration velocity values of i_RMS(α,β,m,ne)_ [mm·s^−1^] sensed on the surface of the Plexiglas trough and on the supporting structure of the vibrating conveyor for the input values α = 10 deg, β = 15 deg, m_m_ = 5.099 kg, and n_ea_ = 769 min^−1^.

Figure 15 gives the mean values of the three times the effective velocity values were measured in the vertical direction (see Table 13, Table 14, Table 15, Table 16, Table 17, Table 18, Table 19, Table 20, Table 21, Table 22, Table 23 and Table 24) and, measured on the frame of the vibrating conveyor model.

## 4. Discussion

The basic idea behind this paper was to verify, whether vibrations measured on the frame of a vibration machine can remotely monitor its working operation and predict its other characteristics.

The aim of this paper has been to demonstrate that it is possible to obtain information about the amount of conveyed material on the trough during the monitored time period from the detected signals measured by vibration acceleration sensors placed on the frame of the vibrating conveyor or sorter. Measured vibration signals detected on the frame of the vibration machine, and processed by the measuring apparatus, allow the machine operator to be informed whether the cylindrical coil springs or rubber springs are in optimum condition or whether they have been damaged or destroyed.

In order to implement any necessary vibration measurements of selected parts of the vibrating machine, it has been necessary to design and construct a model of the vibrating conveyor [32]. Two identical acceleration sensors [37] and one revolution sensor [36] were used to measure the oscillation speed (used at low and medium frequencies f = 10 ÷ 1000 Hz). A DC vibration motor was designed as the oscillation source; its revolutions were controlled by a speed controller [34].

A specific solution was the use of rubber springs, so called silent blocks, to absorb kinetic and acoustic vibrations transmitted to the frame of the vibrating conveyor model. In order to evaluate the measurements, and to verify the correctness of the idea of the possibility of using measured signals by acceleration sensors to monitor the operation of the vibration machine, two types of silent blocks were used, differing in their diameter D [m] and their lengths in the unloaded state H_0_ [m].

The effective values of vibration velocities obtained from the measurements (see Table 1, Table 2, Table 3, Table 4, Table 5, Table 6, Table 7, Table 8, Table 9, Table 10, Table 11, Table 12, Table 13, Table 14, Table 15, Table 16, Table 17, Table 18, Table 19, Table 20, Table 21, Table 22, Table 23 and Table 24) show agreement with the conclusions presented in [8], where the effects of various types of suspension and the ratio of the weight of the body to the weight of the conveyor trough on the forces transmitted to the floor have been analysed.

Limiting the vibration load on the foundations by optimizing (e.g., hull geometry, weight distribution, and spring stiffness) vibration devices is described in [10]. The conclusions presented in the paper [10] are consistent with the results obtained, from Figure 12b it can be seen that the vibrations of the unloaded trough transmitted to the frame of the vibratory conveyor oscillate at a higher value than the trough loaded with the conveyed material.

The results of the measurements of the effective vibration velocity values were realised in this paper for two types of rubber springs, at various values of the trough inclination angle β [deg], throw angle α [deg], i.e., the vector of the excitation force angle F_c_ [N] with respect to the horizontal plane, and load mass m_m_ [kg], i.e., the mass of the material located on the trough surface.

The first set of measurements (see Section 3.1, Section 3.2, Section 3.3 and Section 3.4) was carried out for a rubber spring with the diameter D = 18 mm and length H_0_ = 20 mm, with a stiffness of 54 N·mm^−1^. In Table 1 andTable 12, the measured (repeated three times under the same technical conditions) effective vibration velocity values are recorded. From these measured values, the means and deviations have been calculated according to the Student distribution [41], which are indicated in the last rows of the tables presented in this paper.

The main conclusion from the measurements made can be traced from Figure 12b. The diagram indicates that, with suitably selected spring stiffnesses (for the optimal design of vibratory conveyor spring stiffnesses, it is possible to use the results of [42]) supporting the trough of the vibratory conveyor, it is possible to trace from the analysis of vibration signals (transmitted to the machine frame) generated by sensors whether there is material on the trough [20]. The analysis of Figure 12b clearly indicates that the vibration of the unloaded trough (m_m_ = 0 kg) is higher than the vibration of the loaded trough (m_m_ > 0 kg) of the vibrating conveyor, with a suitably chosen stiffness of the rubber springs s_s_ [N·mm^−1^].

The second set of measurements (see Section 3.5, Section 3.6, Section 3.7 and Section 3.8) was carried out for a rubber spring with diameter D = 25 mm and length H_0_ = 15 mm, with a stiffness of 120 N·mm^−1^. In Table 13, Table 14, Table 15, Table 16, Table 17, Table 18, Table 19, Table 20, Table 21, Table 22, Table 23 and Table 24, the measured effective vibration velocity values are recorded similarly to the previous tables (see Section 3.1, Section 3.2, Section 3.3 and Section 3.4).

The conclusion drawn from the measurements made (Section 3.5, Section 3.6, Section 3.7 and Section 3.8) can be traced from Figure 15, which indicates that with inappropriately chosen spring stiffnesses (s_s_ = 120 N·mm^−1^ compared to the optimal stiffness s_s_ = 54 N·mm^−1^) supporting the trough of the vibratory conveyor, it can be deduced from the analysis of the vibration signals (transmitted to the machine frame) generated by the sensors that the vibrations of the loaded trough (m_m_ > 0 kg) are higher than the vibrations of the loaded trough (m_m_ = 0 kg) of the vibratory conveyor.

The conclusions reached, supported by measurements of vibrations of an oscillating trough, the oscillation of which is excited by a DC asynchronous vibration motor, correspond to the results presented in [21,22,23].

Our aim has been to verify the vibration values measured by the sensors and the results obtained in this paper and to measure the vibration values (monitored by acceleration sensors) on the frame of the vibrating feeder (i.e., a vibrating conveyor with a short trough length, which is designed for mass dosing of the conveyed bulk material); its amplitude and frequency of trough vibration is generated by an electromagnetic vibration exciter. The frequency of the trough vibration, and thus the effective vibration speed values on the frame of this vibrating feeder, will be controlled by changing the frequency of the power grid vibration (maximum magnitude 50 Hz) by the frequency converter of the required electrical power.

## 5. Conclusions

In the presented paper, the tables indicate the effective vibration velocity values obtained by measurements, detected by acceleration sensors, on the trough surface and on the frame of the vibrating conveyor model. The trough of the vibrating conveyor, assembled in one unit from three Plexiglas parts, is supported by four identical rubber springs of two types, which differ in stiffness (spring characteristics). The harmonic oscillation of the trough is produced by a DC asynchronous vibrating motor, the actual revolutions of which are controlled by a thyristor electronic speed controller.

The main objective of the realised signal measurements (which define the magnitude of the vibrations in three mutually perpendicular planes) was to determine whether (with varying input values, namely the angle of throw, the angle of inclination of the trough, and rotor revolutions of the vibrating motor) it is possible to obtain (from the measured magnitudes of the vibrations acting on the frame of the vibrating conveyor) information about the operating characteristics and the mass of material to be conveyed on the trough with respect to the stiffness of the rubber springs supporting the vibrating masses.

Acceleration magnitudes of the effective vibration velocity values measured by sensors have demonstrated and confirmed that if springs of a particular stiffness supporting the vibrating trough are selected appropriately, it is possible to remotely monitor the correct operational operation of the vibratory conveyor and to have the information that the required mass quantity of conveyed/sorted material is on the trough of the vibratory machine.

With the knowledge of the magnitude of the vibrations acting on the frame of a particular vibrating conveyor/sorter (obtained by sensor measurements), with known values of the stiffness of the springs supporting the trough, it is also possible to trace the failure state of their working activities, or to obtain information about the failure or damage of the rubber or steel coil cylindrical springs (used on the vibrating machine).

Signals indicating the magnitude of the vibration values acting on the frame of vibrating conveyors/sorters, transmitted to the control station, allow remote monitoring of the operation of vibrating machines at any time without the need for physical inspection of these devices by authorized persons at a place of their installation.

The obtained data on the magnitudes of the measured signals detected by the vibration sensors allowed us to confirm the correctness of the initial idea that (with appropriately designed machine parts) it is possible to monitor the proper working operation and the failure state of vibrating conveyors under operating conditions.

The current trend towards digitalization and computer-controlled or monitored optimum operation of conveyor handling equipment (including vibratory conveyors and vibratory sorters) is made possible by sensors, measuring equipment and digital signal transmission over any distance.

## Figures and Tables

**Figure 1 sensors-25-02466-f001:**
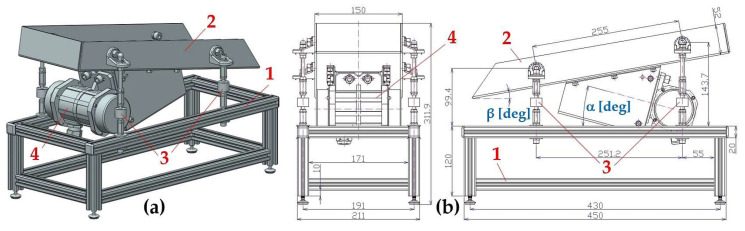
Experimental device created in SolidWorks software (**a**) 3D model; (**b**) 2D sketch; 1—aluminium frame, 2—plastic trough, 3—rubber springs, 4—vibration motor.

**Figure 2 sensors-25-02466-f002:**
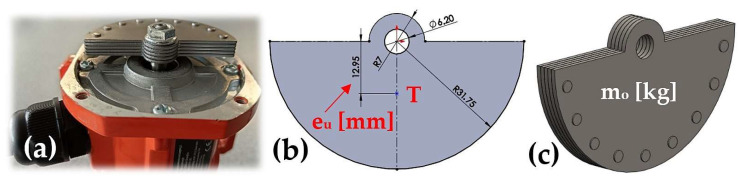
(**a**) VEVOR HY-0.4 asynchronous single-phase motor; (**b**) centre of gravity of the eccentric weight; (**c**) mass of the eccentric weight.

**Figure 3 sensors-25-02466-f003:**
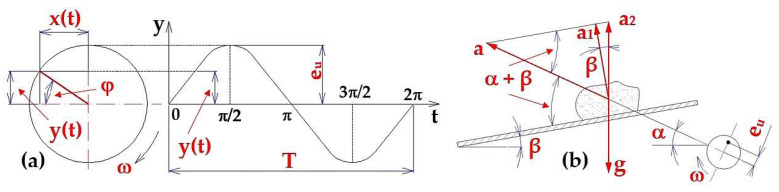
(**a**) Time diagram of harmonic motion; (**b**) acceleration a_i_ [m·s^−2^] of the material grain.

**Figure 4 sensors-25-02466-f004:**
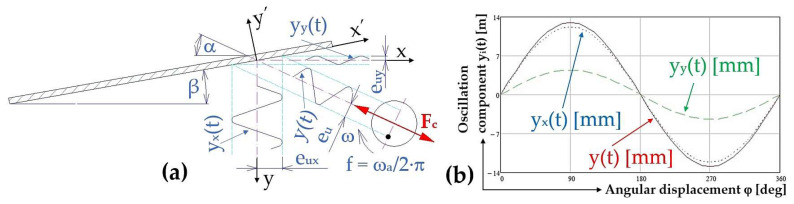
(**a**) Components of the trough oscillation in the direction y_x_(t) [m], and in the direction perpendicular y_y_(t) [m] to the direction of motion; (**b**) components of the trough oscillation y_i_(t) [m] at a throw angle of α = 20 deg and the eccentricity of unbalance of e_u_ = 12.95 mm.

**Figure 5 sensors-25-02466-f005:**
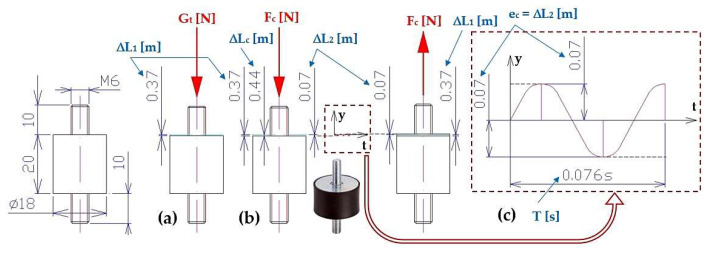
Elastic deformation of the rubber spring (**a**) ΔL_1_ [m] induced by the weight of the system G_s_ [N], (**b**) ΔL_2_ [m] induced by the centrifugal force of the eccentric weights F_c_ [N]; (**c**) harmonic motion of the system.

**Figure 6 sensors-25-02466-f006:**
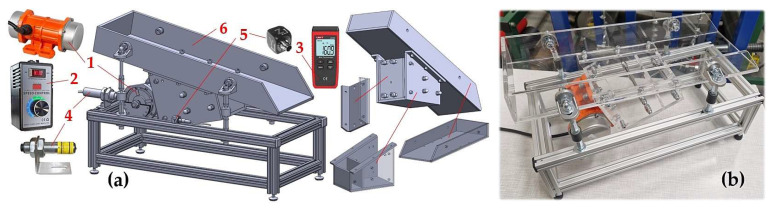
Vibrating conveyor (**a**) control and measuring components; (**b**) realised model with the plastic trough; 1—vibration motor, 2—speed controller, 3—speed sensor, 4—laser sensor, 5—acceleration sensor, 6—plastic trough.

**Figure 7 sensors-25-02466-f007:**
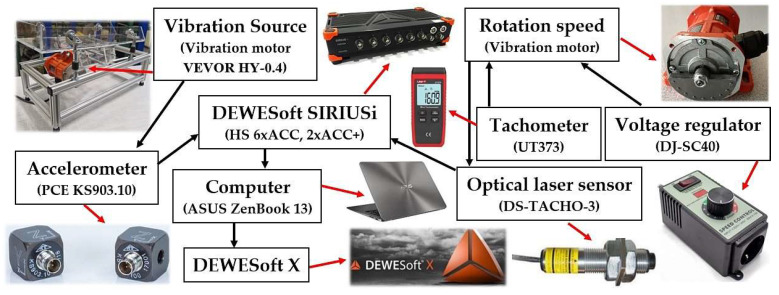
Measurement chain for detecting and recording vibrations generated by rotating eccentric weights on laboratory equipment.

**Figure 8 sensors-25-02466-f008:**
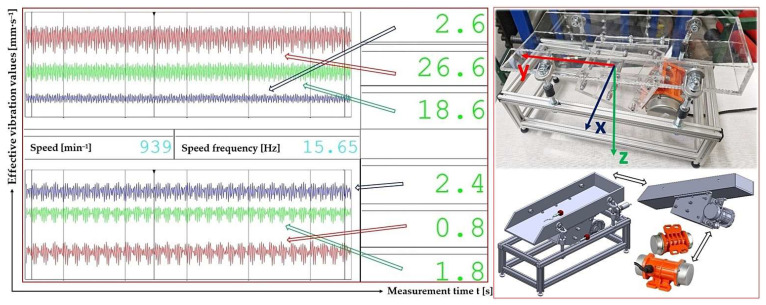
Records of measured effective vibration velocities by acceleration transducers attached to the trough surface and to the frame of the vibrating conveyor model, in the x(**–**), y (**–**), and z-coordinate system (**–**).

**Figure 9 sensors-25-02466-f009:**
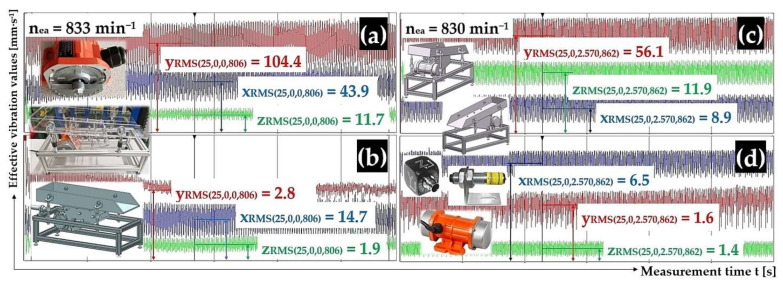
Effective vibration rate values i_RMS(α,β,m,ne)_ [mm·s^−1^] measured (**a**,**c**) on the trough surface, (**b**,**d**) on the frame of the vibrating conveyor model.

**Figure 10 sensors-25-02466-f010:**
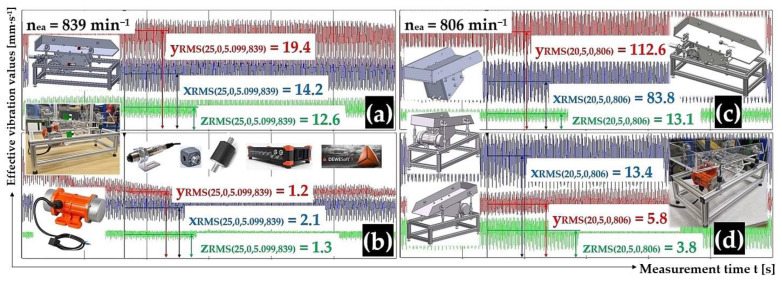
Effective vibration rate values i_RMS(α,β,m,ne)_ [mm·s^−1^] measured (**a**,**c**) on the trough surface and, (**b**,**d**) on the frame of the vibrating conveyor model.

**Figure 11 sensors-25-02466-f011:**
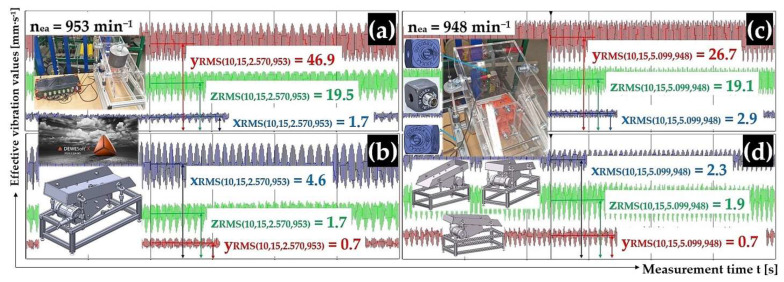
Effective vibration rate values i_RMS(α,β,m,ne)_ [mm·s^−1^] measured (**a**,**c**) on the trough surface, (**b**,**d**) on the frame of the vibrating conveyor model.

**Figure 12 sensors-25-02466-f012:**
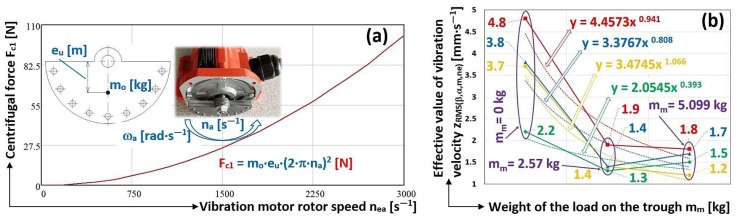
(**a**) The magnitude of the centrifugal force induced by the rotating unbalanced mass; (**b**) the effective values of the vibration velocities in the vertical direction.

**Figure 13 sensors-25-02466-f013:**
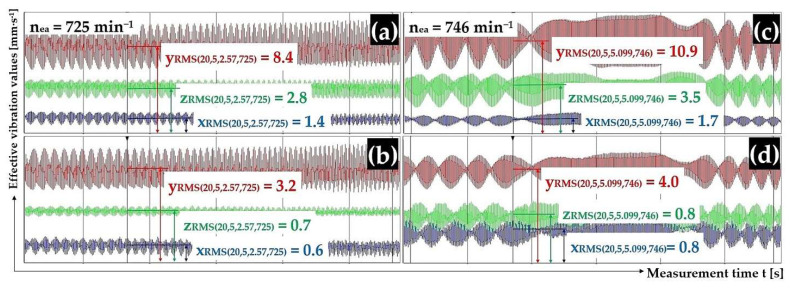
Effective vibration rate values i_RMS(α,β,m,ne)_ [mm·s^−1^] measured (**a**,**c**) on the trough surface, (**b**,**d**) on the frame of the vibrating conveyor model.

**Figure 14 sensors-25-02466-f014:**
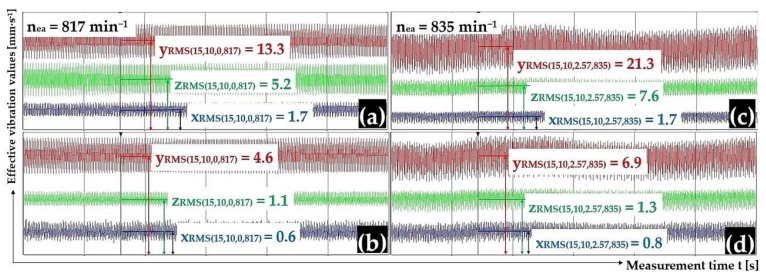
Effective vibration rate values i_RMS(α,β,m,ne)_ [mm·s^−1^] measured (**a**,**c**) on the trough surface, (**b**,**d**) on the frame of the vibrating conveyor model.

**Figure 15 sensors-25-02466-f015:**
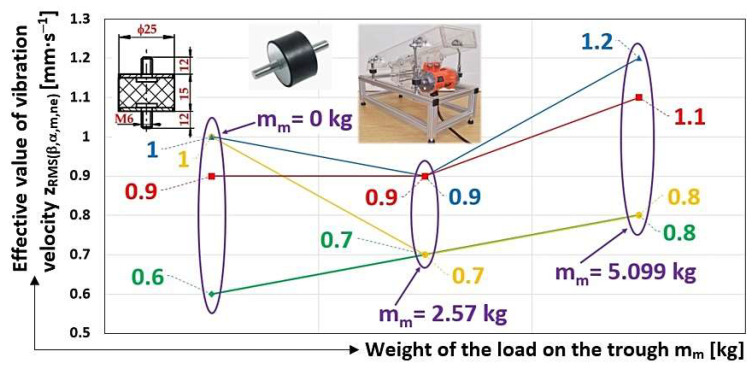
Effective vibration velocity values z_RMS(α,β,m,ne)_ [mm·s^−1^] in the vertical direction, monitored on the vibrating conveyor frame.

**Table 1 sensors-25-02466-t001:** Effective vibration rate values i_RMS(α,β,m,ne)_ [mm·s^−1^] at m_m_ = 0 kg, n_ea_ = 833 min^−1^.

n_ea_	β	α	m_m_	x_RMS(α,β,m,ne)_	y_RMS(α,β,m,ne)_	z_RMS(α,β,m,ne)_	x_RMS(α,β,m,ne)_	y_RMS(α,β,m,ne)_	z_RMS(α,β,m,ne)_
s^−1^	deg		kg	mm·s^−1^			mm·s^−1^		
				trough of vibrating conveyor			d frame of vibrating conveyor		
13.88	0	25	0	43.9 ^1^	104.4 ^1^	11.7 ^1^	14.7 ^2^	2.8 ^2^	1.9 ^2^
				34.3	112.2	12.3	16.0	4.4	2.5
				38.1	108.6	11.9	15.7	3.8	2.1
Σ(i_RMS(α,β,m,ne)_) [mm·s^−1^]				116.3	325.2	35.9	46.4	11.0	6.5
i_RMS(α,β,m,ne)A ±_ κ_5%,3_ [mm·s^−1^]				38.8 ± 8.0	108.4 ± 6.2	12.0 ± 0.5	15.5 ± 1.2	3.7 ± 1.3	2.2 ± 0.5

^1^ See Figure 9a; ^2^ see Figure 9b.

**Table 2 sensors-25-02466-t002:** Effective vibration rate values i_RMS(α,β,m,ne)_ [mm·s^−1^] at m_m_ = 2.57 kg, n_ea_ = 830 min^−1^.

n_ea_	β	α	m_m_	x_RMS(α,β,m,ne)_	y_RMS(α,β,m,ne)_	z_RMS(α,β,m,ne)_	x_RMS(α,β,m,ne)_	y_RMS(α,β,m,ne)_	z_RMS(α,β,m,ne)_
s^−1^	deg		kg	mm·s^−1^			mm·s^−1^		
				trough of vibrating conveyor			frame of vibrating conveyor		
13.83	0	25	2.57	9.8	51.4	10.4	6.1	1.4	1.2
				8.9 ^1^	56.1 ^1^	11.9 ^1^	6.5 ^2^	1.6 ^2^	1.4 ^2^
				9.2	53.3	10.8	6.3	1.4	1.3
Σ(i_RMS(α,β,m,ne)_) [mm·s^−1^]				27.9	160.8	33.1	18.9	4.4	3.9
i_RMS(α,β,m,ne)A ±_ κ_5%,3_ [mm·s^−1^]				9.3 ± 0.8	53.6 ± 3.9	11.0 ± 1.3	6.3 ± 0.3	1.5 ± 0.2	1.3 ± 0.2

^1^ see Figure 9c, ^2^ see Figure 9d.

**Table 3 sensors-25-02466-t003:** Effective vibration rate values i_RMS(α,β,m,ne)_ [mm·s^−1^] at m_m_ = 5.099 kg, n_ea_ = 839 min^−1^.

n_ea_	β	α	m_m_	x_RMS(α,β,m,ne)_	y_RMS(α,β,m,ne)_	z_RMS(α,β,m,ne)_	x_RMS(α,β,m,ne)_	y_RMS(α,β,m,ne)_	z_RMS(α,β,m,ne)_
s^−1^	deg		kg	mm·s^−1^			mm·s^−1^		
				trough of vibrating conveyor			frame of vibrating conveyor		
13.98	0	25	5.099	13.7	24.7	15.3	2.8	1.4	1.7
				14.2 ^1^	19.4 ^1^	12.6 ^1^	2.1 ^2^	1.2 ^2^	1.3 ^2^
				13.9	20.6	13.4	2.6	1.4	1.5
Σ(i_RMS(α,β,m,ne)_) [mm·s^−1^]				41.8	64.7	41.3	7.5	4.0	4.5
i_RMS(α,β,m,ne)A ±_ κ_5%,3_ [mm·s^−1^]				13.9 ± 0.4	21.6 ± 4.9	13.8 ± 2.4	2.5 ± 0.6	1.3 ± 0.2	1.5 ± 0.3

^1^ See Figure 10a; ^2^ see Figure 10b.

**Table 4 sensors-25-02466-t004:** Effective vibration rate values i_RMS(α,β,m,ne)_ [mm·s^−1^] at m_m_ = 0 kg, n_ea_ = 806 min^−1^.

n_ea_	β	α	m_m_	x_RMS(α,β,m,ne)_	y_RMS(α,β,m,ne)_	z_RMS(α,β,m,ne)_	x_RMS(α,β,m,ne)_	y_RMS(α,β,m,ne)_	z_RMS(α,β,m,ne)_
s^−1^	deg		kg	mm·s^−1^			mm·s^−1^		
				trough of vibrating conveyor			frame of vibrating conveyor		
13.43	5	20	0	78.7	125.5	12.8	15.9	6.9	3.6
				83.8 ^1^	112.6 ^1^	13.1 ^1^	13.4 ^2^	5.8 ^2^	3.8 ^2^
				82.2	120.1	12.9	14.5	6.2	3.8
Σ(i_RMS(α,β,m,ne)_) [mm·s^−1^]				244.7	358.2	38.8	43.8	18.9	11.2
i_RMS(α,β,m,ne)A ±_ κ_5%,3_ [mm·s^−1^]				81.6 ± 4.4	119.4 ± 10.6	12.9 ± 0.3	14.6 ± 2.0	6.3 ± 0.9	3.7 ± 0.2

^1^ See Figure 10c; ^2^ see Figure 10d.

**Table 5 sensors-25-02466-t005:** Effective vibration rate values i_RMS(α,β,m,ne)_ [mm·s^−1^] at m_m_ = 2.57 kg, n_ea_ = 862 min^−1^.

n_ea_	β	α	m_m_	x_RMS(α,β,m,ne)_	y_RMS(α,β,m,ne)_	z_RMS(α,β,m,ne)_	x_RMS(α,β,m,ne)_	y_RMS(α,β,m,ne)_	z_RMS(α,β,m,ne)_
s^−1^	deg		kg	mm·s^−1^			mm·s^−1^		
				trough of vibrating conveyor			frame of vibrating conveyor		
14.37	5	20	2.57	21.2	58.5	16.5	6.8	2.5	1.3
				21.1	57.6	17.1	6.7	1.8	1.4
				21.2	58.1	16.9	6.8	2.2	1.4
Σ(i_RMS(α,β,m,ne)_) [mm·s^−1^]				63.5	174.2	50.5	20.3	6.5	4.1
i_RMS(α,β,m,ne)A ±_ κ_5%,3_ [mm·s^−1^]				21.2 ± 0.1	58.1 ± 0.7	16.8 ± 0.5	6.8 ± 0.1	2.2 ± 0.6	1.4 ± 0.1

**Table 6 sensors-25-02466-t006:** Effective vibration rate values i_RMS(α,β,m,ne)_ [mm·s^−1^] at m_m_ = 5.099 kg, n_ea_ = 879 min^−1^.

n_ea_	β	α	m_m_	x_RMS(α,β,m,ne)_	y_RMS(α,β,m,ne)_	z_RMS(α,β,m,ne)_	x_RMS(α,β,m,ne)_	y_RMS(α,β,m,ne)_	z_RMS(α,β,m,ne)_
s^−1^	deg		kg	mm·s^−1^			mm·s^−1^		
				trough of vibrating conveyor			frame of vibrating conveyor		
14.65	5	20	5.099	7.5	27.1	14.8	3.0	1.3	1.2
				9.3	27.0	14.2	3.0	1.6	1.2
				8.6	27.1	14.3	3.1	1.4	1.2
Σ(i_RMS(α,β,m,ne)_) [mm·s^−1^]				25.4	81.2	43.3	9.1	4.3	3.6
i_RMS(α,β,m,ne)A ±_ κ_5%,3_ [mm·s^−1^]				8.5 ± 1.5	27.1 ± 0.1	14.4 ± 0.6	3.0 ± 0.1	1.4 ± 0.3	1.2 ± 0.0

**Table 7 sensors-25-02466-t007:** Effective vibration rate values i_RMS(α,β,m,ne)_ [mm·s^−1^] at m_m_ = 0 kg, n_ea_ = 816 min^−1^.

n_ea_	β	α	m_m_	x_RMS(α,β,m,ne)_	y_RMS(α,β,m,ne)_	z_RMS(α,β,m,ne)_	x_RMS(α,β,m,ne)_	y_RMS(α,β,m,ne)_	z_RMS(α,β,m,ne)_
s^−1^	deg		kg	mm·s^−1^			mm·s^−1^		
				trough of vibrating conveyor			frame of vibrating conveyor		
13.60	10	15	0	3.4	88.4	20.0	17.0	2.8	3.4
				4.0	99.5	21.5	18.8	3.9	4.2
				3.9	97.1	21.3	18.2	3.2	3.8
Σ(i_RMS(α,β,m,ne)_) [mm·s^−1^]				11.3	285.0	62.8	54.0	9.9	11.4
i_RMS(α,β,m,ne)A ±_ κ_5%,3_ [mm·s^−1^]				3.8 ± 0.6	95.0 ± 10.2	20.9 ± 1.4	18.0 ± 1.6	3.3 ± 0.9	3.8 ± 0.6

**Table 8 sensors-25-02466-t008:** Effective vibration rate values i_RMS(α,β,m,ne)_ [mm·s^−1^] at m_m_ = 2.57 kg, n_ea_ = 961 min^−1^.

n_ea_	β	α	m_m_	x_RMS(α,β,m,ne)_	y_RMS(α,β,m,ne)_	z_RMS(α,β,m,ne)_	x_RMS(α,β,m,ne)_	y_RMS(α,β,m,ne)_	z_RMS(α,β,m,ne)_
s^−1^	deg		kg	mm·s^−1^			mm·s^−1^		
				trough of vibrating conveyor			frame of vibrating conveyor		
16.02	10	15	2.57	2.8	58.5	25.4	8.2	1.3	1.6
				2.4	77.3	20.5	11.0	2.0	1.3
				2.6	77.0	23.1	9.6	1.8	1.4
Σ(i_RMS(α,β,m,ne)_) [mm·s^−1^]				7.8	212.8	69.0	28.8	5.1	4.3
i_RMS(α,β,m,ne)A ±_ κ_5%,3_ [mm·s^−1^]				2.6 ± 0.3	70.9 ± 19.3	23.0 ± 3.9	9.6 ± 2.2	1.7 ± 0.6	1.4 ± 0.3

**Table 9 sensors-25-02466-t009:** Effective vibration rate values i_RMS(α,β,m,ne)_ [mm·s^−1^] at m_m_ = 5.099 kg, n_ea_ = 917 min^−1^.

n_ea_	β	α	m_m_	x_RMS(α,β,m,ne)_	y_RMS(α,β,m,ne)_	z_RMS(α,β,m,ne)_	x_RMS(α,β,m,ne)_	y_RMS(α,β,m,ne)_	z_RMS(α,β,m,ne)_
s^−1^	deg		kg	mm·s^−1^			mm·s^−1^		
				trough of vibrating conveyor			frame of vibrating conveyor		
15.28	10	15	5.099	2.3	27.9	17.3	3.5	0.8	1.5
				3.1	27.3	19.8	3.3	0.7	1.9
				2.9	27.6	19.2	3.5	0.8	1.7
Σ(i_RMS(α,β,m,ne)_) [mm·s^−1^]				8.3	82.8	56.3	10.3	2.3	5.1
i_RMS(α,β,m,ne)A ±_ κ_5%,3_ [mm·s^−1^]				2.8 ± 0.7	27.6 ± 0.5	18.8 ± 2.3	3.4 ± 0.2	0.8 ± 0.1	1.7 ± 0.3

**Table 10 sensors-25-02466-t010:** Effective vibration rate values i_RMS(α,β,m,ne)_ [mm·s^−1^] at m_m_ = 0 kg, n_ea_ = 748 min^−1^.

n_ea_	β	α	m_m_	x_RMS(α,β,m,ne)_	y_RMS(α,β,m,ne)_	z_RMS(α,β,m,ne)_	x_RMS(α,β,m,ne)_	y_RMS(α,β,m,ne)_	z_RMS(α,β,m,ne)_
s^−1^	deg		kg	mm·s^−1^			mm·s^−1^		
				trough of vibrating conveyor			frame of vibrating conveyor		
12.47	15	10	0	6.1	122.7	23.3	23.0	4.7	5.0
				6.1	125.7	23.6	21.3	4.1	4.7
				6.1	124.2	23.5	21.1	4.3	4.8
Σ(i_RMS(α,β,m,ne)_) [mm·s^−1^]				18.3	372.6	70.4	65.4	13.1	14.5
i_RMS(α,β,m,ne)A ±_ κ_5%,3_ [mm·s^−1^]				6.1 ± 0.0	124.2 ± 2.3	23.5 ± 0.3	21.8 ± 1.9	4.4 ± 0.5	4.8 ± 0.3

**Table 11 sensors-25-02466-t011:** Effective vibration rate values i_RMS(α,β,m,ne)_ [mm·s^−1^] at m_m_ = 2.57 kg, n_ea_ = 853 min^−1^.

n_ea_	β	α	m_m_	x_RMS(α,β,m,ne)_	y_RMS(α,β,m,ne)_	z_RMS(α,β,m,ne)_	x_RMS(α,β,m,ne)_	y_RMS(α,β,m,ne)_	z_RMS(α,β,m,ne)_
s^−1^	deg		kg	mm·s^−1^			mm·s^−1^		
				trough of vibrating conveyor			frame of vibrating conveyor		
15.88	15	10	2.57	1.7 ^1^	46.9 ^1^	19.5 ^1^	4.6 ^2^	0.7 ^2^	1.7 ^2^
				1.8	44.6	21.2	4.2	0.8	2.0
				1.7	46.0	19.6	4.4	0.7	1.9
Σ(i_RMS(α,β,m,ne)_) [mm·s^−1^]				5.2	137.5	60.3	13.2	2.2	5.6
i_RMS(α,β,m,ne)A ±_ κ_5%,3_ [mm·s^−1^]				1.7 ± 0.1	45.8 ± 1.9	20.1 ± 1.7	4.4 ± 0.3	0.7 ± 0.1	1.9 ± 0.3

^1^ See Figure 11a; ^2^ see Figure 11b.

**Table 12 sensors-25-02466-t012:** Effective vibration rate values i_RMS(α,β,m,ne)_ [mm·s^−1^] at m_m_ = 5.099 kg, n_ea_ = 948 min^−1^.

n_ea_	β	α	m_m_	x_RMS(α,β,m,ne)_	y_RMS(α,β,m,ne)_	z_RMS(α,β,m,ne)_	x_RMS(α,β,m,ne)_	y_RMS(α,β,m,ne)_	z_RMS(α,β,m,ne)_
s^−1^	deg		kg	mm·s^−1^			mm·s^−1^		
				trough of vibrating conveyor			frame of vibrating conveyor		
15.80	15	10	5.099	2.9 ^1^	26.7 ^1^	19.1 ^1^	2.3 ^2^	0.7 ^2^	1.9 ^2^
				2.6	26.6	18.6	2.4	0.8	1.8
				2.7	26.6	18.9	2.4	0.8	1.8
Σ(i_RMS(α,β,m,ne)_) [mm·s^−1^]				8.2	79.9	56.6	7.1	2.3	5.5
i_RMS(α,β,m,ne)A ±_ κ_5%,3_ [mm·s^−1^]				2.7 ± 0.3	26.6 ± 0.1	18.9 ± 0.4	2.4 ± 0.1	0.8 ± 0.1	1.8 ± 0.1

^1^ See Figure 11c; ^2^ see Figure 11d.

**Table 13 sensors-25-02466-t013:** Effective vibration rate values i_RMS(α,β,m,ne)_ [mm·s^−1^] at m_m_ = 0 kg, n_ea_ = 761 min^−1^.

n_ea_	β	α	m_m_	x_RMS(α,β,m,ne)_	y_RMS(α,β,m,ne)_	z_RMS(α,β,m,ne)_	x_RMS(α,β,m,ne)_	y_RMS(α,β,m,ne)_	z_RMS(α,β,m,ne)_
s^−1^	deg		kg	mm·s^−1^			mm·s^−1^		
				trough of vibrating conveyor			frame of vibrating conveyor		
12.68	0	25	0	1.4	7.3	2.4	0.8	3.3	0.6
				1.1	6.8	2.2	0.7	3.0	0.6
				1.3	7.1	2.3	0.8	3.1	0.6
Σ(i_RMS(α,β,m,ne)_) [mm·s^−1^]				3.8	21.2	6.9	2.3	9.4	1.8
i_RMS(α,β,m,ne)A ±_ κ_5%,3_ [mm·s^−1^]				1.3 ± 0.3.0	7.1 ± 0.4	2.3 ± 0.2	0.8 ± 0.1	3.1 ± 0.3	0.6 ± 0.0

**Table 14 sensors-25-02466-t014:** Effective vibration rate values i_RMS(α,β,m,ne)_ [mm·s^−1^] at m_m_ = 2.57 kg, n_ea_ = 830 min^−1^.

n_ea_	β	α	m_m_	x_RMS(α,β,m,ne)_	y_RMS(α,β,m,ne)_	z_RMS(α,β,m,ne)_	x_RMS(α,β,m,ne)_	y_RMS(α,β,m,ne)_	z_RMS(α,β,m,ne)_
s^−1^	deg		kg	mm·s^−1^			mm·s^−1^		
				trough of vibrating conveyor			frame of vibrating conveyor		
12.45	0	25	2.57	1.1	7.7	2.6	0.8	3.6	0.7
				1.1	7.6	2.4	0.8	3.4	0.7
				1.1	7.7	2.6	0.8	3.5	0.7
Σ(i_RMS(α,β,m,ne)_) [mm·s^−1^]				3.3	23.0	7.6	2.4	10.5	2.1
i_RMS(α,β,m,ne)A ±_ κ_5%,3_ [mm·s^−1^]				1.1 ± 0.0	7.7 ± 0.1	2.5 ± 0.2	0.8 ± 0.0	3.5 ± 0.2	0.7 ± 0.0

**Table 15 sensors-25-02466-t015:** Effective vibration rate values i_RMS(α,β,m,ne)_ [mm·s^−1^] at m_m_ = 5.099 kg, n_ea_ = 750 min^−1^.

n_ea_	β	α	m_m_	x_RMS(α,β,m,ne)_	y_RMS(α,β,m,ne)_	z_RMS(α,β,m,ne)_	x_RMS(α,β,m,ne)_	y_RMS(α,β,m,ne)_	z_RMS(α,β,m,ne)_
s^−1^	deg		kg	mm·s^−1^			mm·s^−1^		
				trough of vibrating conveyor			frame of vibrating conveyor		
13.98	0	25	5.099	1.2	8.7	2.5	0.8	3.8	0.8
				1.4	9.9	2.6	0.9	4.4	0.7
				1.4	9.4	2.6	0.9	4.1	0.8
Σ(i_RMS(α,β,m,ne)_) [mm·s^−1^]				4.0	28.0	7.7	2.6	12.3	2.3
i_RMS(α,β,m,ne)A ±_ κ_5%,3_ [mm·s^−1^]				1.3 ± 0.2	9.3 ± 1.0	2.6 ± 0.1	0.9 ± 0.1	4.1 ± 0.5	0.8 ± 0.1

**Table 16 sensors-25-02466-t016:** Effective vibration rate values i_RMS(α,β,m,ne)_ [mm·s^−1^] at m_m_ = 0 kg, n_ea_ = 847 min^−1^.

n_ea_	β	α	m_m_	x_RMS(α,β,m,ne)_	y_RMS(α,β,m,ne)_	z_RMS(α,β,m,ne)_	x_RMS(α,β,m,ne)_	y_RMS(α,β,m,ne)_	z_RMS(α,β,m,ne)_
s^−1^	deg		kg	mm·s^−1^			mm·s^−1^		
				trough of vibrating conveyor			frame of vibrating conveyor		
14.12	5	20	0	1.4	10.7	3.8	0.6	4.8	0.9
				1.6	12.7	4.0	0.7	5.3	1.1
				1.6	12.3	3.8	0.7	5.1	1.1
Σ(i_RMS(α,β,m,ne)_) [mm·s^−1^]				4.6	35.7	11.6	2.0	15.2	3.1
i_RMS(α,β,m,ne)A ±_ κ_5%,3_ [mm·s^−1^]				1.5. ± 0.2	11.9 ± 1.9	3.9 ± 0.2	0.7 ± 0.1	5.1 ± 0.4	1.0 ± 0.2

**Table 17 sensors-25-02466-t017:** Effective vibration rate values i_RMS(α,β,m,ne)_ [mm·s^−1^] at m_m_ = 2.57 kg, n_ea_ = 828 min^−1^.

n_ea_	β	α	m_m_	x_RMS(α,β,m,ne)_	y_RMS(α,β,m,ne)_	z_RMS(α,β,m,ne)_	x_RMS(α,β,m,ne)_	y_RMS(α,β,m,ne)_	z_RMS(α,β,m,ne)_
s^−1^	deg		kg	mm·s^−1^			mm·s^−1^		
				trough of vibrating conveyor			frame of vibrating conveyor		
12.08	5	20	2.57	1.4 ^1^	8.4 ^1^	2.8 ^1^	0.6 ^2^	3.2 ^2^	0.7 ^2^
				1.7	9.3	3.0	0.7	3.6	0.8
				1.5	9.2	3.3	0.7	3.8	0.7
Σ(i_RMS(α,β,m,ne)_) [mm·s^−1^]				4.6	26.9	9.1	2.0	10.6	2.2
i_RMS(α,β,m,ne)A ±_ κ_5%,3_ [mm·s^−1^]				1.5 ± 0.3	9.0 ± 0.9	3.0 ± 0.4	0.7 ± 0.1	3.5 ± 0.5	0.7 ± 0.1

^1^ see Figure 13a, ^2^ see Figure 13b.

**Table 18 sensors-25-02466-t018:** Effective vibration rate values i_RMS(α,β,m,ne)_ [mm·s^−1^] at m_m_ = 5.099 kg, n_ea_ = 746 min^−1^.

n_ea_	β	α	m_m_	x_RMS(α,β,m,ne)_	y_RMS(α,β,m,ne)_	z_RMS(α,β,m,ne)_	x_RMS(α,β,m,ne)_	y_RMS(α,β,m,ne)_	z_RMS(α,β,m,ne)_
s^−1^	deg		kg	mm·s^−1^			mm·s^−1^		
				trough of vibrating conveyor			frame of vibrating conveyor		
12.43	5	20	5.099	1.7	9.3	3.0	0.7	3.6	0.8
				1.7 ^1^	10.9 ^1^	3.5 ^1^	0.8 ^2^	4.0 ^2^	0.8 ^2^
				1.8	9.9	3.1	0.8	3.3	0.8
Σ(i_RMS(α,β,m,ne)_) [mm·s^−1^]				5.2	20.1	9.6	2.3	10.9	2.4
i_RMS(α,β,m,ne)A ±_ κ_5%,3_ [mm·s^−1^]				1.7 ± 0.1	10.0 ± 1.3	3.2 ± 0.5	0.8 ± 0.1	3.6 ± 0.6	0.8 ± 0.0

^1^ See Figure 13c; ^2^ see Figure 13d.

**Table 19 sensors-25-02466-t019:** Effective vibration rate values i_RMS(α,β,m,ne)_ [mm·s^−1^] at m_m_ = 0 kg, n_ea_ = 817 min^−1^.

n_ea_	β	α	m_m_	x_RMS(α,β,m,ne)_	y_RMS(α,β,m,ne)_	z_RMS(α,β,m,ne)_	x_RMS(α,β,m,ne)_	y_RMS(α,β,m,ne)_	z_RMS(α,β,m,ne)_
s^−1^	deg		kg	mm·s^−1^			mm·s^−1^		
				trough of vibrating conveyor			frame of vibrating conveyor		
13.62	10	15	0	1.7 ^1^	13.3 ^1^	5.2 ^1^	0.6 ^2^	4.6 ^2^	1.1 ^2^
				1.7	12.3	4.3	0.6	4.6	0.8
				1.8	12.6	4.8	0.6	4.7	1.0
Σ(i_RMS(α,β,m,ne)_) [mm·s^−1^]				5.2	38.2	14.3	1.8	13.9	2.9
i_RMS(α,β,m,ne)A ±_ κ_5%,3_ [mm·s^−1^]				1.7 ± 0.1	12.7 ± 0.9	4.8 ± 0.7	0.6 ± 0.0	4.6 ± 0.1	1.0 ± 0.3

^1^ See Figure 14a; ^2^ see Figure 14b.

**Table 20 sensors-25-02466-t020:** Effective vibration rate values i_RMS(α,β,m,ne)_ [mm·s^−1^] at m_m_ = 2.57 kg, n_ea_ = 835 min^−1^.

n_ea_	β	α	m_m_	x_RMS(α,β,m,ne)_	y_RMS(α,β,m,ne)_	z_RMS(α,β,m,ne)_	x_RMS(α,β,m,ne)_	y_RMS(α,β,m,ne)_	z_RMS(α,β,m,ne)_
s^−1^	deg		kg	mm·s^−1^			mm·s^−1^		
				trough of vibrating conveyor			frame of vibrating conveyor		
13.92	10	15	2.57	1.7 ^1^	21.3 ^1^	7.6 ^1^	0.8 ^2^	6.9 ^2^	1.3 ^2^
				2.0	22.1	7.5	0.7	7.3	1.1
				1.8	21.6	7.5	0.8	7.1	1.3
Σ(i_RMS(α,β,m,ne)_) [mm·s^−1^]				5.5	65.0	22.6	2.3	21.3	3.7
i_RMS(α,β,m,ne)A ±_ κ_5%,3_ [mm·s^−1^]				1.8 ± 0.3	21.7 ± 0.7	7.5 ± 0.1	0.8 ± 0.1	7.1 ± 0.3	1.2 ± 0.2

^1^ see Figure 14c, ^2^ see Figure 14d.

**Table 21 sensors-25-02466-t021:** Effective vibration rate values i_RMS(α,β,m,ne)_ [mm·s^−1^] at m_m_ = 5.099 kg, n_ea_ = 779 min^−1^.

n_ea_	β	α	m_m_	x_RMS(α,β,m,ne)_	y_RMS(α,β,m,ne)_	z_RMS(α,β,m,ne)_	x_RMS(α,β,m,ne)_	y_RMS(α,β,m,ne)_	z_RMS(α,β,m,ne)_
s^−1^	deg		kg	mm·s^−1^			mm·s^−1^		
				trough of vibrating conveyor			frame of vibrating conveyor		
12.98	10	15	5.099	1.2	25.7	9.5	0.8	8.3	1.2
				1.0	23.6	8.3	0.7	7.6	1.1
				1.0	26.5	9.5	0.9	8.5	1.2
Σ(i_RMS(α,β,m,ne)_) [mm·s^−1^]				3.2	75.8	27.3	2.4	24.4	3.5
i_RMS(α,β,m,ne)A ±_ κ_5%,3_ [mm·s^−1^]				1.1 ± 0.2	25.3 ± 2.6	9.1 ± 1.2	0.8 ± 0.2	8.1 ± 0.8	1.2 ± 0.1

**Table 22 sensors-25-02466-t022:** Effective vibration rate values i_RMS(α,β,m,ne)_ [mm·s^−1^] at m_m_ = 0 kg, n_ea_ = 849 min^−1^.

n_ea_	β	α	m_m_	x_RMS(α,β,m,ne)_	y_RMS(α,β,m,ne)_	z_RMS(α,β,m,ne)_	x_RMS(α,β,m,ne)_	y_RMS(α,β,m,ne)_	z_RMS(α,β,m,ne)_
s^−1^	deg		kg	mm·s^−1^			mm·s^−1^		
				trough of vibrating conveyor			frame of vibrating conveyor		
14.15	15	10	0	2.4	15.8	6.4	0.8	5.9	1.3
				2.0	10.6	4.3	0.7	4.4	0.8
				1.6	9.8	3.8	0.6	3.9	0.7
Σ(i_RMS(α,β,m,ne)_) [mm·s^−1^]				6.0	36.2	14.5	2.1	14.2	2.8
i_RMS(α,β,m,ne)A ±_ κ_5%,3_ [mm·s^−1^]				2.0 ± 0.6	12.1 ± 5.8	4.8 ± 2.4	0.7 ± 0.2	4.7 ± 1.8	0.9 ± 0.6

**Table 23 sensors-25-02466-t023:** Effective vibration rate values i_RMS(α,β,m,ne)_ [mm·s^−1^] at m_m_ = 2.57 kg, n_ea_ = 751 min^−1^.

n_ea_	β	α	m_m_	x_RMS(α,β,m,ne)_	y_RMS(α,β,m,ne)_	z_RMS(α,β,m,ne)_	x_RMS(α,β,m,ne)_	y_RMS(α,β,m,ne)_	z_RMS(α,β,m,ne)_
s^−1^	deg		kg	mm·s^−1^			mm·s^−1^		
				trough of vibrating conveyor			frame of vibrating conveyor		
12.52	15	10	2.57	2.4	15.2	5.9	0.6	5.3	0.9
				2.8	11.9	4.9	0.5	4.3	0.7
				3.2	11.5	5.1	0.6	4.2	0.8
Σ(i_RMS(α,β,m,ne)_) [mm·s^−1^]				8.4	38.6	15.9	1.7	13.8	2.4
i_RMS(α,β,m,ne)A ±_ κ_5%,3_ [mm·s^−1^]				2.8 ± 0.6	12.9 ± 3.6	5.3 ± 0.9	0.6 ± 0.1	4.6 ± 1.1	0.8 ± 0.2

**Table 24 sensors-25-02466-t024:** Effective vibration rate values i_RMS(α,β,m,ne)_ [mm·s^−1^] at m_m_ = 5.099 kg, n_ea_ = 769 min^−1^.

n_ea_	β	α	m_m_	x_RMS(α,β,m,ne)_	y_RMS(α,β,m,ne)_	z_RMS(α,β,m,ne)_	x_RMS(α,β,m,ne)_	y_RMS(α,β,m,ne)_	z_RMS(α,β,m,ne)_
s^−1^	deg		kg	mm·s^−1^			mm·s^−1^		
				trough of vibrating conveyor			frame of vibrating conveyor		
12.82	15	10	5.099	3.1	23.8	9.8	0.3	8.4	1.1
				3.2	24.4	10.1	0.4	8.5	1.1
				3.0	23.6	9.6	0.4	8.2	1.2
Σ(i_RMS(α,β,m,ne)_) [mm·s^−1^]				9.3	71.8	29.5	1.1	25.1	3.4
i_RMS(α,β,m,ne)A ±_ κ_5%,3_ [mm·s^−1^]				3.1 ± 0.2	23.9 ± 0.7	9.8 ± 0.4	0.4 ± 0.1	8.4 ± 0.3	1.1 ± 0.1

## Data Availability

Measured data of effective vibration speed values i_RMS(α,β,m,ne)_ [mm·s^−1^], listed from Table 1, Table 2, Table 3, Table 4, Table 5, Table 6, Table 7, Table 8, Table 9, Table 10, Table 11, Table 12, Table 13, Table 14, Table 15, Table 16, Table 17, Table 18, Table 19, Table 20, Table 21, Table 22, Table 23 and Table 24 and processed using DEWESoft X software, can be sent in case of interest, by prior written agreement, in *.XLSX (Microsoft Excel) format.

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
