# Peer review of "Sensor Monitoring of Conveyor Working Operation with Oscillating Trough Movement"

_sensors, 2025, doi:10.3390/s25082466_

Round 1

Reviewer 1 Report

Comments and Suggestions for Authors

This study investigated the vibration monitoring method for vibrating conveyors during operation. By utilizing acceleration sensors to measure the vibration velocities of both the trough and frame, it analyzed the effects of rubber spring stiffness on vibration transmission. The research outcomes provide theoretical guidance for the design and operation-maintenance of vibration equipment.

However, the manuscript has some unclear or incomplete statements that need to be revised and improved prior to possible publication. The comments are as follows.

  1. The abstract requires further refinement to concisely highlight the research objectives, key methodologies, and principal conclusions of the study.  
  2. The content in lines 568-570 on page 21 appears to be tangential to the core research focus and recommend their removal.  
  3. Consider including a comparative figure to visually demonstrate the impact of spring stiffness on vibration velocity.  
  4. Clarification is needed regarding the discrepancy in spring stiffness parameters: the second measurement group's spring stiffness (ss = 120 N·mm⁻¹) cited on page 22 line 635 conflicts with the value (ss = 1834 N·mm⁻¹) specified on page 16 line 480.  
  5. This research direction warrants further exploration, particularly regarding fault localization in large-scale vibrating conveyors with multiple silent blocks. The positioning methodology for defective silent blocks should be incorporated into fault diagnosis systems, with reference to the following literature for technical implementation:Study on Sensor Network Optimization for MS/AE Monitoring System Using Fisher Information and Improved Encoding Framework. Joint Inversion of AE/MS Sources and Velocity with Full Measurements and Residual Estimation.A robust triaxial localization method of AE source using refraction path.

Author Response

Dear Reviewer.

Thank you very much for your valuable advice and comments.

I greatly appreciate your opinion and respectfully thank you for your comments.

Sincerely, Leopold Hrabovský.

Reviewer 2 Report

Comments and Suggestions for Authors

Dear author,

This manuscript is of great significance to the structural vibration analysis and related experiments of vibrating conveyors. It introduces a vibrating conveyor test bench and explores how to collect vibration acceleration data that meets the actual operating parameters of rubber springs under laboratory conditions. However, further review and processing are needed.

  1. The introduction of the article makes a broad summary of past work, but the research objectives are vague, the innovations are not clear, and the contribution of this article is not pointed out;
  2. It is not clear whether the test bench parameters are designed with reference to actual equipment or obtained through calculations, and the design lacks theoretical basis, such as spring stiffness selection and motor speed setting;
  3. The results section only repeats the experimental results without summarizing the research contributions. The discussion section is too brief and only compares with the data in related papers. The data measured by the test bench should try to reflect the equipment data under actual industrial conditions, and the engineering significance of the experimental results is not deeply explored;
  4. The conclusion is too simple, only repeating the experimental results, for example, the vibration amplitude of the test bench frame reflects the relationship between different structural parameters on material quality and rubber spring stiffness, which is not summarized, and the research objectives or methods are not fully described;
  5. The article mentioned that the experimental model is used for related research on rubber spring damage information monitoring, but the article did not mention any relevant data;
  6. The vibrating conveyor is an equipment for transporting small pieces, granules, and powdery materials. Will the large block load selected in the experiment affect the data collected in the experiment? Obviously, this has a great impact on the vibration of the trough;
  7. Two types of rubber springs (54 N/mm and 120 N/mm) were selected in the experiment. Do these two stiffness types cover the typical range of rubber springs in actual industrial applications?
  8. The paper mentioned that the maximum allowable compression of the rubber spring is 5.0 mm (54 N/mm) and 3.75 mm (120 N/mm), but the compression in the experiment is only 0.44 mm and 0.2 mm. Does it mean that the conclusion of the paper is only applicable to a small deformation range?
  9. The load mass in the experiment is set to 0 kg, 2.57 kg, and 5.099 kg. Do these values ​​have practical reference significance? For example, is it proportional to the rated load of the conveyor? Or is it calculated based on typical values ​​of material density?

Author Response

(The authors gave the same response as above.)

Reviewer 3 Report

Comments and Suggestions for Authors

This interesting paper with application in industry presents measured vibration magnitudes on the trough surface and on the frame of a laboratory model of a vibrating conveyor, using acceleration sensors. The paper has extensive experimental results. The literature review is substantial and the provided references are up-to-date and relevant for the topic. The paper is overall well written, in a clear and rigorous style, the theoretical and technical part is presented in detail. The included figures are very clear and explain the obtained results.

Please revise the following issues in order to improve the paper quality:

1) page 5 line 186: "throw angle alpha ... can be quantified according to (1)" (?) - equation (1) does not contain any angle "alpha", please check and revise       

2) page 14 line 429: "Centrifugal force Fc1 [N], see Figure 12(a), increasing exponentially with increasing angular velocity" (?) - as is well known, and also as shows equation (1), the centrifugal force is proportional to the square of the angular velocity, it does not varies exponentially with it. Please revise and correct.
Therefore, the plot in figure 12(a) should be parabolic, not exponential.    

3) There are many tables (1 through 24) which display experimentally measured effective vibration rate values in various working conditions (set of parameters); as stated in conclusions, "the tables indicate the effective vibration velocity values obtained by measurements, detected by acceleration sensors"; however, no clear conclusion is drawn from all these experimental data. Please synthesize a general conclusion regarding all these measured data.

4) page 14 line 434: What is the  meaning of "refracted curves"? 

5) REFERENCES
The references are not written uniformly; most titles are in simple case (like [1] etc.), others in title case (like [6], [7], [17], [18], [20], [21], [23], [32]); 
Journal names should be written in Italics, according to the template.
Please verify and revise, according to the available manuscript template.

Comments on the Quality of English Language

GRAMMAR should be revised throughout the manuscript; make the following corrections as suggested:    
page 1 line 17: subjected to the deailed -> subjected to a detailed          
page 2 line 54: is beneficial in that that if we determine -> is beneficial in that if we determine              
page 2 line 55: if we determine at least one of the determining quantities (?) -> if we determine at least one of the fundamental quantities        
page 2 line 59: The article analyses the impacts ... transmitted to the ground are analysed. (?) -> The article analyses the impacts ... transmitted to the ground.       
page 2 line 64, 65: presents a spatial model ... by two inertial vibrators is presented in this article. -> presents a spatial model ... by two inertial vibrators.     
page 2 line 66: The results of analyses of the effect of the layout of vibrators on the operation of the conveyor -> The results of analyses of the effect of vibrators layout on the conveyor operation           
page 2 line 79: "the composite signal with two sinusoidal components of double frequency" - unclear, reformulate     
page 2 line 84: using the integrated Runge-Kutta methods -> using the Runge-Kutta numerical integration methods          
page 3 line 101: or rms value (RMS) -> or RMS value           
page 3 line 106: with adynamic damper -> with a dynamic damper (?)          
page 3 line 106: presented in this paper [16] -> presented in paper [16]          
page 3 line 113: vibration insuation system  -> vibration insulation system       
page 3 line 119: "present a mathematical description ... was derived, and parameters of this model were estimated" (?) - reformulate    
page 3 line 139: conveyors, which principle -> conveyors, whose principle          
page 4 line 155: Howerver -> However         
page 4 line 158: For the design of the experimental device, see Figure 1, has been designed at -> The experimental device (see Figure 1) has been designed at            
page 7 line 252: hmotnosti -> mass           
page 7 line 257: "single-phase motor of vibration motor 1" - unclear, reformulate     
page 7 line 257: Speed ... were controlled -> Speed ... was controlled                
page 8 line 316: see chapters 3.1 to 3.8 -> see subsections 3.1 to 3.8                   
page 15 line 439: theeffective values -> the effective values                     
page 15 line 441: chap. 3.1 to chap. 3.4 -> subsections 3.1 to 3.4             
page 15 line 445: Chapters 3.5 to 3.8 -> Subsections 3.5 to 3.8                 
page 15 line 467: chap. 3.5 (see Table 13) to chap. 3.8 ->  subsection 3.5 (see Table 13) to subsection 3.8          
page 21 line 578: "to ask whether vibrations ... can remotely monitor" - unclear, reformulate                 
page 22 line 625: "v porovnání s optimální tuhostí" - please translate to English                        

Author Response

Dear Reviewer.

Thank you very much for your valuable advice and comments. Thank you also for the errors found in the English language.

I greatly appreciate your opinion and respectfully thank you for your comments.

I have made corrections to the text of the contribution according to your recommendation. I have marked all changes in green in the manuscript.

Thank you very much for correcting the shortcomings in the English language.

With respect, Leopold Hrabovský.

Reviewer 4 Report

Comments and Suggestions for Authors

The article investigates the dynamics of a vibration system with an unbalanced rotor, used in a conveyor drive system. The relevance of the work lies in the wide application of vibrating conveyors, screens, mixers in the oil refining, construction and mining industries. The article is written in a strict scientific style, its theoretical provisions are fundamentally substantiated, and its practical applicability is confirmed by experimental stand.

The graphs, calculation schemes and tables in the work are used logically and allow for a more qualitative disclosure of the essence of the article. However, I would like to add one remark to improve the quality of the article. In my opinion, figures 8-11, 13-14 have very small captions and are difficult to read. The scale of the figures should be increased.

A positive feature of the article is the extensive discussion section before the conclusion, which interprets the main ideas, provisions and results of the work. The list of references is formatted correctly and contains references to works in reputable journals.

Author Response

Dear Reviewer.

Thank you for your positive review of our manuscript. We greatly appreciate your opinion and comments on the illegibility of the captions in Figures 9-11; 13 and 14. I have corrected the captions for these figures. The correction can be verified in the submitted manuscript.

Best regards. Sincerely, Leopold Hrabovský.
